# Navigating Sparsities in High-Dimensional Linear Contextual Bandits

## Abstract

High-dimensional linear contextual bandit problems remain a significant challenge due to the curse of dimensionality. Existing methods typically consider either the model parameters to be sparse or the eigenvalues of context covariance matrices to be (approximately) sparse, lacking general applicability due to the rigidity of conventional reward estimators. To overcome this limitation, a powerful pointwise estimator is introduced in this work that adaptively navigates both kinds of sparsity. Based on this pointwise estimator, a novel algorithm, termed HOPE, is proposed. Theoretical analyses demonstrate that HOPE not only achieves improved regret bounds in previously discussed homogeneous settings (i.e., considering only one type of sparsity) but also, for the first time, efficiently handles two new challenging heterogeneous settings (i.e., considering a mixture of two types of sparsity), highlighting its flexibility and generality. Experiments corroborate the superiority of HOPE over existing methods across various scenarios.

## 1 Introduction

The contextual bandit framework has emerged as a powerful tool for decision-making applications (Chu et al., 2011; Agrawal & Goyal, 2013), where an agent selects arms based on contextual information and receives corresponding rewards. The low-dimensional setting, where the context dimension is small compared with the time horizon, is considerably well-understood through many pioneer works (Abbasi-Yadkori et al., 2011; Agrawal & Goyal, 2013; Lattimore & Szepesvári, 2020; Hao et al., 2020). The high-dimensional setting (Bastani & Bayati, 2020; Negahban et al., 2012; Hao et al., 2020; Li et al., 2022; Kim & Paik, 2019; Oh et al., 2021; Ren & Zhou, 2024; Qian et al., 2024; Cai et al., 2023; Han et al., 2025; Shi et al., 2023), where the context dimension is comparable with or even larger than the time horizon, is yet under-explored.

Especially, in high-dimensional settings, the complexity introduced by numerous contextual features poses significant challenges, commonly referred to as the curse of dimensionality, and resulting in vacuous results (Abbasi-Yadkori et al., 2011; Chu et al., 2011) from low-dimensional approaches. As this curse intuitively cannot be lifted in general scenarios, existing works mostly focused on leveraging additional structural considerations to bypass it. Two mostly considered structures are both regarding sparsity: (I) assuming the *model parameters are sparse*, where Lasso-based algorithms have been extensively studied for identifying relevant context features, achieving sublinear regret bounds (Li et al., 2022; Bastani & Bayati, 2020; Hao et al., 2020) and (II) assuming *the covariance matrices of context distributions have (approximately) sparse eigenvalues*, where recent work by Komiyama & Imaizumi (2024) employs the ridgeless least-squares (RDL) estimator (Bartlett et al., 2020), also achieving sublinear regrets in various cases. However, as shown in Fig. 1, these methods face limitations, as they can only handle one type of sparsity, restricting their general applicability.

This lack of flexibility in previous works originates from the rigidity of their adopted estimators, i.e., Lasso and RDL. This work introduces a powerful PointWise Estimator (PWE) based on the recent breakthrough in Zhao et al. (2023). Based on PWE, a novel algorithm, HOPE (High-dimensional linear cOntextual bandits with Pointwise Estimator), is proposed. HOPE follows the explore-then-commit (ETC) scheme with PWE as the main estimator after the exploration phase. The detailed contributions of HOPE are further summarized in the following:

• *Novelty.* Existing high-dimensional bandit methods rely on *sparsity-specific* estimators (e.g., Lasso or RDL) and can exploit only one structural assumption at a time. To the best of our knowledge, HOPE is the first bandit algorithm that is capable of adaptively navigating both types of sparsity (i.e., the model parameter and the eigenvalues of context covariance matrices) at the same time via PWE. Building on this, we introduce and rigorously study two challenging *heterogeneous* scenarios: (i) each arm exhibits both sparsity types simultaneously; (ii) different arms follow different sparsity types.

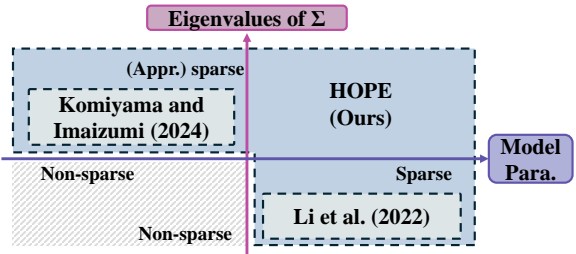

Figure 1: Applicability of previous works and HOPE, where the third quadrant, marked gray, is in general non-solvable.

• *Theory.* Comprehensive theoretical analyses have been established for HOPE, providing a thorough demonstration of its effectiveness and efficiency. One general regret guarantee is provided. Based on it, four different scenarios are further discussed. We first prove that in the two *homogeneous* scenarios with one type of sparsity, HOPE matches the theory guarantees from prior work. More importantly, in the other two challenging *heterogeneous* scenarios, HOPE behaves in a theoretically efficient manner while previous works fail.

• *Practicality.* Our experimental results further validate the theoretical advances across all four scenarios, showcasing HOPE's flexibility and superior performance.

## 2 PROBLEM FORMULATION

This work considers a linear contextual bandit problem involving $K$ arms and $T$ rounds, with a particular focus on high-dimensional scenarios (Komiyama & Imaizumi, 2024; Bastani & Bayati, 2020; Li et al., 2022).

**Contexts.** At each round $t \in [T]$, an arm context $\boldsymbol{x}_t^{(i)} \in \mathbb{R}^p$ is received for each arm $i \in [K]$. Without loss of generality, $\boldsymbol{x}_t^{(i)}$ is considered to be sampled from a $p$-dimensional zero-mean distribution $\mathcal{P}_i$, as in Komiyama & Imaizumi (2024). To ease the discussion, the context distribution $\mathcal{P}_i$ is considered to be a zero-mean Gaussian one with a covariance matrix denoted as $\boldsymbol{\Sigma}^{(i)} = \mathbb{E}[\boldsymbol{x}_t^{(i)}(\boldsymbol{x}_t^{(i)})^\top] \in \mathbb{R}^{p \times p}$, i.e., $\mathcal{N}(0, \boldsymbol{\Sigma}^{(i)})$. Note that this assumption does not restrict generality; our results can be readily extended to sub-Gaussian distributions with non-zero means by incorporating minor adjustments.

It is further assumed that for each arm $i \in [K]$, the sampling of $\boldsymbol{x}_t^{(i)}$ is independent across rounds, i.e., the contexts of one arm in two different rounds $t, t'$ are independent, while within the same round, the contexts of different arms, i.e., $\{\boldsymbol{x}_t^{(i)} : i \in [K]\}$, can be correlated.

**Rewards.** Based on the arm contexts $\{\boldsymbol{X}_t^{(i)} : i \in [K]\}$, the agent chooses an arm $i(t) \in [K]$, and subsequently observes a reward $y_t^{(i(t))}$ that follows a linear model: $y_t^{(i(t))} = \mu_t^{(i(t))} + \varepsilon(t)$, where the expected reward $\mu_t^{(i)}$ is parameterized as $\mu_t^{(i)} := \langle \boldsymbol{x}_t^{(i)}, \boldsymbol{\theta}^{(i)} \rangle$, with $\{\boldsymbol{\theta}^{(i)} \in \mathbb{R}^p : i \in [K]\}$ as unknown model parameters, while $\varepsilon(t)$ captures an independent zero-mean noise with its variance denoted as $\sigma^2 > 0$. We assume that each $\boldsymbol{\theta}^{(i)}$ is bounded $\|\boldsymbol{\theta}^{(i)}\|_2 \leq \theta_{\max}$.

**The Design Objective.** The optimal arm at round $t$ is defined as $i^*(t) := \arg\max_{i \in [K]} \mu_t^{(i)}$. Following the canonical MAB research (Lai & Robbins, 1985; Auer et al., 2002), the design objective is to maximize the expected cumulative rewards of $T$ rounds, which is captured by minimizing the expected regret $R(T)$ defined as $R(T) := \mathbb{E}\left[\sum_{t=1}^T \mu_t^{(i^*(t))} - \mu_t^{(i(t))}\right]$. It is noted that the above expectation is taken with respect to the randomness of the context distributions and potentially the arm selections.

**The High-dimensional Setting.** As mentioned in Sec. 1, unlike the majority of works in linear (contextual) bandit (Chu et al., 2011; Abbasi-Yadkori et al., 2011; Agrawal & Goyal, 2013), where the feature dimension is (implicitly) assumed to be moderate compared with the horizon $T$, i.e., $p \ll T$ (referred to as the low-dimensional setting). This work, instead, focuses on the high-dimensional

setting (Komiyama & Imaizumi, 2024; Li et al., 2022; Bastani & Bayati, 2020; Hao et al., 2020) with the feature dimension $p$ at least on the same order as the budget $T$, i.e., $p \gtrsim T$.[1]

The high-dimensional setting is widely recognized as notoriously challenging, as highlighted in Sec. 1. With the canonical low-dimensional regret guarantees of order $\tilde{O}(\text{poly}(p)\sqrt{T})$ becoming vacuous (where $\text{poly}(\cdot)$ denoting a polynomial term of the input), the general target in the high-dimensional setting is to obtain a regret that is $p$-independent, i.e., not scaling with the feature dimension, while maintains sublinear in $T$. However, this task in general is non-achievable without further structural information, as there certainly lacks sufficient data to faithfully estimate the unknown system parameters in the worst case.

Current studies primarily focus on two types of sparse structures: one related to model parameters (Li et al., 2022; Bastani & Bayati, 2020; Wang et al., 2018; Hao et al., 2020) and the other concerning the covariances of arm contexts (Komiyama & Imaizumi, 2024).

• *Sparsity of Model Parameters.* The model parameters, i.e., $\{\boldsymbol{\theta}^{(i)} = [\theta_1^{(i)}, \cdots, \theta_p^{(i)}] : i \in [K]\}$, exhibit sparsity, meaning that only a few parameters are non-zero. We denote the support set for arm $i$ as $\mathcal{S}_0^{(i)} := \{j \in [p] : \theta_j^{(i)} \neq 0\}$, with $s_0^{(i)} := |\mathcal{S}_0^{(i)}|$. In this case, it is commonly considered that $s_0 := \max_{i \in [K]} s_0^{(i)} \ll p$, i.e., the effective dimension is much lower than the model dimension.

• *(Approximate) Sparsity of Context Covariance Eigenvalues.* [2] The properties of the covariance matrices of arm contexts, i.e., $\{\boldsymbol{\Sigma}^{(i)} : i \in [K]\}$, can also be considered. One particular case is that the covariance matrix approximately exhibits sparsity in its eigenvalues, i.e., only a few eigenvalues are significantly larger than the others. With a rigorous quantification detailed in Sec. 6, we provide two examples to illustrate this structure following Komiyama & Imaizumi (2024). We refer to this structure as "*sparse eigenvalues of* $\boldsymbol{\Sigma}$" in the later presentations.

**Example 1.** *Two examples of the approximate sparsity of context covariance eigenvalues (Komiyama & Imaizumi, 2024):*

*(A)* $\lambda_k(\boldsymbol{\Sigma}^{(i)}) = k^{-(1+1/T^a)}$ *for all* $k \in [p]$ *when* $a \in (0,1)$;

*(B) let* $p = O(T^c)$, $\lambda_k(\boldsymbol{\Sigma}^{(i)}) = k^{-b}$ *for all* $k \in [p]$ *when* $b \in (0,1)$ *and* $c \in (1, 1/1-b)$.

Our work mostly follows the problem formulation in Komiyama & Imaizumi (2024) and is motivated to provide a unified solution that can leverage these two structures in a more adaptive manner.

**Remark 1.** It is noted that the settings studied in previous works (Li et al., 2022; Bastani & Bayati, 2020; Wang et al., 2018; Hao et al., 2020; Komiyama & Imaizumi, 2024) are not identical to each other. A detailed comparison of these settings is provided in App. B.

## 3 THE PREVIOUSLY ADOPTED ESTIMATORS

The key challenge in the high-dimensional linear bandit lies in effectively estimating the arm reward $\mu_t^{(i)}$ given its context $\boldsymbol{x}_t^{(i)}$ under the high-dimensional structure. Different kinds of estimators have been adopted in existing works. In this section, we provide an overview of these previously considered estimators, especially Lasso and RDL[3], which have their advantages in certain regimes but lack general flexibility.

To facilitate the discussion, we focus on one arm $i$ as an example and consider that a dataset containing $N$ pairs of independently generated arm contexts and rewards, denoted as $\{\boldsymbol{x}_\tau^{(i)}, y_\tau^{(i)} : \tau \in [N]\}$, which can be imagined as collected from an exploration phase, e.g., following the explore-then-commit (ETC) procedure as in (Li et al., 2022; Komiyama & Imaizumi, 2024) and the later proposed

---

[1] With the number of unknown model parameters being $Kp$, one problem can be considered as high-dimensional if $Kp \gtrsim T$. We use the convention $p \gtrsim T$ here and in the later discussions.

[2] We adopt the terminology in Zhao et al. (2023). It describes the same spectral-sparsity phenomenon commonly discussed under the notions of eigenvalue decay and small effective rank (cf. Bartlett et al. (2020)).

[3] Another commonly adopted estimator in linear bandits (Abbasi-Yadkori et al., 2011) is the ridge estimator (Hoerl & Kennard, 1970). As its power is mostly confined to the low-dimensional setting while this work is focused on the high-dimensional setting, it is not discussed here.

HOPE algorithm. For convenience, we further denote $\mathbf{X}^{(i)} := [\boldsymbol{x}_1^{(i)}, \cdots, \boldsymbol{x}_N^{(i)}]^\top \in \mathbb{R}^{N \times p}$ and $\boldsymbol{y}^{(i)} := [y_1^{(i)}, \cdots, y_N^{(i)}] \in \mathbb{R}^N$.

**Lasso.** The Lasso estimator (Tibshirani, 1996) minimizes the sum of squared residuals with an $l_1$-norm regularization: $\hat{\boldsymbol{\theta}}_{\mathrm{Lasso}}^{(i)} = \arg\min_{\boldsymbol{\theta}} \left\{ \|\boldsymbol{y}^{(i)} - \mathbf{X}^{(i)}\boldsymbol{\theta}\|_2^2 + \lambda\|\boldsymbol{\theta}\|_1 \right\}$, where $\lambda$ is the regularization parameter. It can be observed that Lasso encourages sparsity in the estimates; thus, it is adopted for high-dimensional sparse linear bandits (Li et al., 2022; Hao et al., 2020; Bastani & Bayati, 2020).

**RDL.** The ridgeless least squares (RDL) estimator (Bartlett et al., 2020) leverages benign overfitting and is given by: $\hat{\boldsymbol{\theta}}_{\mathrm{RDL}}^{(i)} = \arg\min_{\boldsymbol{\theta}} \left\{ \|\boldsymbol{\theta}\|_2 \big| \|\boldsymbol{y}^{(i)} - \mathbf{X}^{(i)}\boldsymbol{\theta}\|_2^2 = \min_{\boldsymbol{\beta}} \|\boldsymbol{y}^{(i)} - \mathbf{X}^{(i)}\boldsymbol{\beta}\|_2^2 \right\} = (\mathbf{X}^{(i)})^\top (\mathbf{X}^{(i)}(\mathbf{X}^{(i)})^\top)^{-1}\boldsymbol{y}^{(i)}$, (Komiyama & Imaizumi, 2024) adopt RDL in high-dimensional bandit problem as it leverages the approximately sparse eigenvalues of $\boldsymbol{\Sigma}^{(i)}$.

**Limitations.** We highlight two concrete limitations: (i) *No joint exploitation when structures coexist.* When both structures are present, neither Lasso nor RDL can exploit them concurrently; each leverages at most one and leaves the other unutilized, which leads to suboptimal statistical rates and regret guarantees in such mixed-structure problems. (ii) *Homogeneity assumption across arms.* Both methods are typically analyzed under a single, uniform structural assumption. They do not accommodate heterogeneous scenarios where different arms follow different structures, nor do they provide a principled mechanism to combine information across such heterogeneous arms.

## 4 A POWERFUL POINTWISE ESTIMATOR

The aforementioned limitations of previously adopted estimators motivate us to introduce a recently proposed PointWise Estimator (PWE) (Zhao et al., 2023), which serves as the foundation for the HOPE algorithm presented in Sec. 5. An overview of PWE is provided in the following, illustrating its suitability for high-dimensional linear contextual bandit problems. The setting from Sec. 3 is inherited that there is an i.i.d. dataset $\{\boldsymbol{x}_\tau^{(i)}, y_\tau^{(i)} : \tau \in [2N]\}$ for arm $i$, based on which we describe the estimation process of $\mu_t^{(i)}$ with one received context $\boldsymbol{x}_t^{(i)}$. Here, we consider the dataset size as $2N$ to facilitate the discussion. In particular, to ensure independence between the preparation step in Sec. 4.1 and the other steps, we split the dataset into two halves: $\{\boldsymbol{x}_\tau^{(i)}, y_\tau^{(i)} : \tau \in [N]\}$ and $\{\boldsymbol{x}_\tau^{(i)}, y_\tau^{(i)} : \tau \in [N+1, 2N]\}$.

### 4.1 ESTIMATING THE SUPPORT SET AND THE INITIAL ESTIMATOR

With the first half of the dataset, several preparation steps are performed to facilitate further estimations. First, the support set $\mathcal{S}_0^{(i)}$ is estimated. This process can be conducted by varying variable selection techniques (Fan & Lv, 2008; Tibshirani, 1996; Candes et al., 2018), such as the standard approach of using a Lasso estimator, with more detailed in App. C.1. We denote the estimated support set as $\mathcal{S}_1^{(i)} \subseteq [p]$ with $s_1^{(i)} := |\mathcal{S}_1^{(i)}|$.

Then, In this process, an initial estimator of $\boldsymbol{\theta}^{(i)}$, denoted as $\hat{\boldsymbol{\theta}}^{(i)}$ is needed, which in this work are considered to be obtained via either Lasso or RDL with the second half of the dataset.

With the estimated support set $\mathcal{S}_1^{(i)}$, the arm contexts in the second half of the dataset, i.e., $\{\boldsymbol{x}_\tau^{(i)} : \tau \in [N+1, 2N]\}$, and the received $\boldsymbol{x}_t^{(i)}$ can be truncated to their sub-vectors with elements at positions contained in $\mathcal{S}_1^{(i)}$. We slightly abuse $\mathbf{X}^{(i)}$ to denote $\left[\boldsymbol{x}_{N+1}^{(i)}[\mathcal{S}_1^{(i)}], \cdots, \boldsymbol{x}_{2N}^{(i)}[\mathcal{S}_1^{(i)}]\right]^\top \in \mathbb{R}^{N \times s_1^{(i)}}$, while using $\boldsymbol{x}_t^{(i)}$ and $\boldsymbol{\theta}^{(i)}$ to refer to the truncated $\boldsymbol{x}_t^{(i)}[\mathcal{S}_1^{(i)}]$ and $\boldsymbol{\theta}^{(i)}[\mathcal{S}_1^{(i)}]$.

### 4.2 TRANSFORMING THE LINEAR MODEL

First, we denote $\mathbf{P}_t^{(i)} := \boldsymbol{x}_t^{(i)}(\boldsymbol{x}_t^{(i)})^\top / \|\boldsymbol{x}_t^{(i)}\|_2^2$ as the projection matrix on the space spanned by $\boldsymbol{x}_t^{(i)}$ and $\mathbf{Q}_t^{(i)} := \mathbf{I}_{s_1^{(i)}} - \mathbf{P}_t^{(i)}$ as the projection matrix on the complementary space. The following relationship can be formulated $\mathbf{X}^{(i)}\boldsymbol{\theta}^{(i)} = \mathbf{X}^{(i)}\mathbf{P}_t^{(i)}\boldsymbol{\theta}^{(i)} + \mathbf{X}^{(i)}\mathbf{Q}_t^{(i)}\boldsymbol{\theta}^{(i)} = \sqrt{N}\alpha_t^{(i)}\boldsymbol{z}_t^{(i)} + \sqrt{N}\boldsymbol{\zeta}_t^{(i)}$,

with the following definitions:

$$\alpha_t^{(i)} := \frac{\|\mathbf{X}^{(i)}\boldsymbol{x}_t^{(i)}\|_2}{\sqrt{N}\|\boldsymbol{x}_t^{(i)}\|_2^2}(\boldsymbol{x}_t^{(i)})^\top\boldsymbol{\theta}^{(i)} = \frac{\|\mathbf{X}^{(i)}\boldsymbol{x}_t^{(i)}\|_2}{\sqrt{N}\|\boldsymbol{x}_t^{(i)}\|_2^2}\cdot\mu_t^{(i)} \in \mathbb{R},$$

$$\boldsymbol{z}_t^{(i)} := \frac{\mathbf{X}^{(i)}\boldsymbol{x}_t^{(i)}}{\|\mathbf{X}^{(i)}\boldsymbol{x}_t^{(i)}\|_2} \in \mathbb{R}^N, \quad \boldsymbol{\zeta}_t^{(i)} := \frac{\mathbf{X}^{(i)}\mathbf{Q}_t^{(i)}\boldsymbol{\theta}^{(i)}}{\sqrt{N}} \in \mathbb{R}^N.$$

It can be noted that estimating $\mu_t^{(i)}$ is equivalent to estimating $\alpha_t^{(i)}$, as the scaling parameter can be directly computed from $(\mathbf{X}^{(i)}, \boldsymbol{x}_t^{(i)})$.

Based on this relationship, we can get that

$$\boldsymbol{y}^{(i)} = \mathbf{X}^{(i)}\boldsymbol{\theta}^{(i)} + \boldsymbol{\varepsilon}^{(i)} = \sqrt{N}\alpha_t^{(i)}\boldsymbol{z}_t^{(i)} + \sqrt{N}\boldsymbol{\zeta}_t^{(i)} + \boldsymbol{\varepsilon}^{(i)}. \tag{1}$$

**Remark 2.** Note that this transformation allows PWE to have $N+1$ unknown parameters, instead of the $p$ dimensions in $\boldsymbol{\theta}^{(i)}$, where $p \gtrsim T > N$ in the high-dimensional settings.

### 4.3 SPARSIFYING THE NUISANCE VECTOR

As $\boldsymbol{\zeta}_t^{(i)}$ is in general a non-sparse vector, Eqn. (1) can be observed to have $N+1$ unknown parameters (with the target $\alpha_t^{(i)}$ and the other $N$ nuisances from $\boldsymbol{\zeta}_t^{(i)}$). However, there are only $N$ conditions from the $N$ samples in the dataset. To enable the estimation, we construct an invertible matrix $\boldsymbol{\Gamma}_t^{(i)} \in \mathbb{R}^{N\times N}$, which transforms the nuisance vector into a sparse representation. The specific construction of $\boldsymbol{\Gamma}_t^{(i)}$ is detailed in App. C.2.

In particular, it can be considered that $\sqrt{N}\boldsymbol{\zeta}_t^{(i)} = (\sqrt{N}\boldsymbol{\Gamma}_t^{(i)})((\boldsymbol{\Gamma}_t^{(i)})^{-1}\boldsymbol{\zeta}_t^{(i)}) = \sqrt{N}\boldsymbol{\Gamma}_t^{(i)}\boldsymbol{\xi}_t^{(i)}$, where $\boldsymbol{\xi}_t^{(i)} := (\boldsymbol{\Gamma}_t^{(i)})^{-1}\boldsymbol{\zeta}_t^{(i)} \in \mathbb{R}^N$ is the transformed nuisance vector. With a properly chosen $\boldsymbol{\Gamma}_t^{(i)}$, it can be obtained that $\boldsymbol{\xi}_t^{(i)}$ is (approximately) sparse.

Combining with Eqn. 1, it holds that $\boldsymbol{y}^{(i)} = \sqrt{N}\alpha_t^{(i)}\boldsymbol{z}_t^{(i)} + \sqrt{N}\boldsymbol{\Gamma}_t^{(i)}\boldsymbol{\xi}_t^{(i)} + \boldsymbol{\varepsilon}^{(i)} = \mathbf{Z}_t^{(i)}\boldsymbol{\beta}_t^{(i)} + \boldsymbol{\varepsilon}^{(i)}$, where $\mathbf{Z}_t^{(i)} = \sqrt{N}\cdot[\boldsymbol{z}_t^{(i)}, \boldsymbol{\Gamma}_t^{(i)}] \in \mathbb{R}^{N\times(N+1)}$, and $\boldsymbol{\beta}_t^{(i)} = [\alpha_t^{(i)}, (\boldsymbol{\xi}_t^{(i)})^\top]^\top \in \mathbb{R}^{N+1}$.

We note that the $N+1$ dimensional vector $\boldsymbol{\beta}_t^{(i)}$ is the target to be solved. Due to the sparsity in $\boldsymbol{\xi}_t^{(i)}$, although there are only $N$ conditions, it is still solvable.

### 4.4 THE OVERALL PWE PROCEDURE

We consider minimizing the following Lasso objective with a regularization parameter $\lambda_t^{(i)}$:

$$\hat{\boldsymbol{\beta}}_t^{(i)} = \operatorname*{arg\,min}_{\boldsymbol{\beta}\in\mathbb{R}^{N+1}} \frac{1}{N}\|\boldsymbol{y}^{(i)} - \mathbf{Z}_t^{(i)}\boldsymbol{\beta}\|_2^2 + \lambda_t^{(i)}\|\boldsymbol{\beta}\|_1. \tag{2}$$

The target $\hat{\alpha}_t^{(i)}$ can be further obtained as $\hat{\boldsymbol{\beta}}_t^{(i)} = [\hat{\alpha}_t^{(i)}, \hat{\boldsymbol{\xi}}_t^{(i)\top}]^\top$, and finally, we have:

$$\hat{\mu}_t^{(i)} = \hat{\alpha}_t^{(i)}\cdot\frac{\sqrt{N}\|\boldsymbol{x}_t^{(i)}\|_2^2}{\|\mathbf{X}^{(i)}\boldsymbol{x}_t^{(i)}\|_2}. \tag{3}$$

The entire procedure of PWE is summarized in Alg. 1.

## 5 THE HOPE ALGORITHM

With the PWE estimator introduced in Sec. 4, we propose our High-dimensional linear cOntextual bandits with Pointwise Estimator algorithm, abbreviated as HOPE. This algorithm is based on the well-known Explore-then-Commit (ETC) scheme, which starts with an exploration phase and is followed by an exploitation (or known as commitment) phase. Its effectiveness in high-dimensional linear bandit problems has also been demonstrated in previous studies (Hao et al., 2020; Li et al., 2022; Komiyama & Imaizumi, 2024).

**Algorithm 1** PWE for arm $i$

**Input:** Dataset $\{\boldsymbol{x}_\tau^{(i)}, y_\tau^{(i)} : \tau \in [2N]\}$, context $\boldsymbol{x}_t^{(i)}$, regularization parameter $\lambda_t^{(i)}$
1: With the first half of the dataset, obtain the estimated support set $\mathcal{S}_1^{(i)}$, the initial estimator $\hat{\boldsymbol{\theta}}^{(i)}$
2: Truncate $\mathbf{X}^{(i)}$ and $\boldsymbol{x}_t^{(i)}$ with $\mathcal{S}_1^{(i)}$
3: Solve $\hat{\boldsymbol{\beta}}_t^{(i)} = [\hat{\alpha}_t^{(i)}, \hat{\boldsymbol{\xi}}_t^{(i)\top}]^\top$ from Eqn. (2)
4: Obtain $\hat{\mu}_t^{(i)}$ from Eqn. (3)
**Output:** Estimate $\hat{\mu}_t^{(i)}$

**Algorithm 2** HOPE

**Input:** Exploration length $T_0 = 2NK$.
1: Explore all arms in a round-robin manner for $T_0$ rounds and obtain $\{\boldsymbol{x}_\tau^{(i)}, y_\tau^{(i)} : \tau \in [2N]\}_{i \in [K]}$
2: **for** $t = T_0 + 1, ..., T$ **do**
3:    Observe $\{\boldsymbol{x}_t^{(i)} : i \in [K]\}$.
4:    Get PWE estimator $\{\hat{\mu}_t^{(i)} : i \in [K]\}$ from Alg. 1 with datasets from the exploration phase
5:    Choose arm $i(t) \leftarrow \arg\max_{i \in [K]} \hat{\mu}_t^{(i)}$
6:    Receive reward $y_t^{(I(t))}$
7: **end for**

Table 1: Regret Comparisons in Different Scenarios. **Parameters**: $K$ denotes the number of arms; $s_0$ denotes the support size of model parameters; $p$ denotes the feature dimension; $T$ denotes the time horizon; $\alpha, a, b,$ and $c$ are example-dependent constants.

| Scenario | Reference | Regret |
|---|---|---|
| Sparse Model Param. | Li et al. (2022) | $O(K^{\frac{1}{3}} s_0^{\frac{1}{3}} T^{\frac{2}{3}} \text{polylog}(pT))$ |
| | Proposition 1 | $O(K^{\frac{1}{3}} s_0^{\frac{1}{3}} T^{\frac{2}{3}} \text{polylog}(T))$ |
| Sparse Eigenvalues of $\boldsymbol{\Sigma}$ | Komiyama & Imaizumi (2024) | $\tilde{O}(K^{\frac{2}{3}} T^{\max\{\frac{2+a}{3}, 1 - \frac{a}{2}\}})$ |
| with Example 1(A) | Proposition 2 | $\tilde{O}(\max\{K^{\frac{1}{2}} p^{\frac{1}{2T\alpha}} T^{\frac{a+2}{4}}, K^{\frac{1}{3}} p^{\frac{2}{3T\alpha}} T^{\frac{2-a}{3}}\})$ |
| Sparse Eigenvalues of $\boldsymbol{\Sigma}$ | Komiyama & Imaizumi (2024) | $\tilde{O}(K^{\frac{2}{3}} T^{\frac{2}{3} + \frac{c(1-b)}{3}})$ |
| with Example 1(B) | Proposition 5 | $\tilde{O}\left(\min\left\{K^{\frac{1}{2}} T^{\frac{1}{2} + \frac{c(2-b)}{4}}, K^{\frac{1}{2}} T^{\frac{1}{2} + \frac{3c(1-b)}{4}}\right\}\right)$ |
| Both Sparsities | Proposition 3 | $\tilde{O}(K^{\frac{1}{3}} M^{\frac{2}{3}} T^{\frac{2}{3}})$ |
| Mixed Sparsities with Example 1(A) | Proposition 4 | $\tilde{O}(\max\{K^{\frac{1}{3}} s_0^{\frac{1}{3}} T^{\frac{2}{3}}, K^{\frac{1}{2}} p^{\frac{1}{2T\alpha}} T^{\frac{a+2}{4}}, K^{\frac{1}{3}} p^{\frac{2}{3T\alpha}} T^{\frac{2-a}{3}}\})$ |

Here, we consider that the exploration phase lasts $T_0 = 2NK < T$ rounds, where all available arms are selected by a round-robin manner (i.e., in turn) for $2N$ times. Then, after the exploration, each arm $i \in [K]$ is associated with a dataset $\{\boldsymbol{x}_\tau^{(i)}, y_\tau^{(i)} : \tau \in [2N]\}$, with a slightly abused notation $\tau$ denoting the $\tau$-th time arm $i$ being pulled. Also, since the arms are uniformly explored, these pairs are i.i.d with each other, as considered in Secs. 3 and 4.

With these datasets, the algorithm proceeds to the exploitation phase. At each time step $t$, estimates $\{\hat{\mu}_t^{(1)}, \cdots, \hat{\mu}_t^{(K)}\}$ can be obtained based on the given arm contexts $\{\boldsymbol{x}_t^{(1)}, \cdots, \boldsymbol{x}_t^{(K)}\}$ through the PWE estimator described in Sec. 4. Then, the empirically optimal arm is selected as $i(t) = \arg\max_{k \in [K]} \hat{\mu}_t^{(k)}$. The HOPE algorithm is presented in Alg. 2.

# 6 THEORETICAL ANALYSIS

We provide a comprehensive set of theoretical results on the performance of HOPE, highlighting its effectiveness and flexibility. A summary of our results and the comparison with existing works (Li et al., 2022; Komiyama & Imaizumi, 2024) under different cases can be found in Table 1. We first list a few assumptions in the following.

**Assumption 1.** *There exists positive constants $c_1, c_2, c_3$ and $c_4$ such that for each arm $i \in [K]$, the covariance matrix $\boldsymbol{\Sigma}^{(i)}$ and the model parameter $\boldsymbol{\theta}^{(i)}$ satisfy the following conditions: (A) $\text{var}((\boldsymbol{\theta}^{(i)})^\top \boldsymbol{x}^{(i)}) = (\boldsymbol{\theta}^{(i)})^\top \boldsymbol{\Sigma}^{(i)} \boldsymbol{\theta}^{(i)} \in [c_1, c_2]$; (B) the largest eigenvalue $\lambda_1(\boldsymbol{\Sigma}^{(i)}) \leq c_3 p / \log T$; (C) $\|\boldsymbol{\Sigma}^{(i)}\|_F / \|\boldsymbol{\Sigma}^{(i)}\|_2 > c_4 \log T$.*

We note that these assumptions are either standard or moderate. Especially, Condition (A) considers that the variance of the expected reward over the context distribution is properly bounded. Conditions (B) and (C) are common requirements for high-dimensional covariance matrices.[4]

---

[4]The theoretical analysis is expressed in $\log N$, but we write $\log T$ for clarity. As $N$ is data-dependent and satisfies $N \leq T$, the relation $\log N \leq \log T$ ensures that the $\log T$ formulation subsumes the required $\log N$ bounds and avoids forward references to $N$.

To derive universal regret guarantees that hold for a broad class of initial estimators—irrespective of the accuracy of support estimation or the duration of the exploration phase—we introduce Assumps. 2 and 3. These impose mild conditions on the initial estimator and support estimates; they are *not required* for Props. 1- 4 but they are essential for the general bound in Thm. 1. Under conventional regularity conditions, common estimators such as Lasso and RDL satisfy Assumps. 2 and 3 with high probability. This ensures the wide applicability of our theoretical results. A rigorous verification of these technical conditions, including their validity in typical problem settings, is provided in App. G.

**Assumption 2.** *With the same $c_1$ as in Assump. 1, for all arm $i \in [K]$, the initial estimator $\hat{\boldsymbol{\theta}}^{(i)}$ satisfies that $|\hat{\boldsymbol{\theta}}^{(i)\top}\boldsymbol{\Sigma}^{(i)}\hat{\boldsymbol{\theta}}^{(i)} - \boldsymbol{\theta}^{(i)\top}\boldsymbol{\Sigma}^{(i)}\boldsymbol{\theta}^{(i)}| \leq c_1/2$.*

**Assumption 3.** *For all arm $i \in [K]$, $\mathcal{S}_1^{(i)}$ satisfies that $\mathcal{S}_0^{(i)} \subseteq \mathcal{S}_1^{(i)}$ and $|\mathcal{S}_1^{(i)}| \leq C_1|\mathcal{S}_0^{(i)}|$.*

The following theorem provides a general regret guarantee.

**Theorem 1.** *Under Assumps. 1, 2 and 3, with an exploration phase lasting $T_0 = 2NK \leq T$ steps and $\lambda_N^{(i)} \asymp \sigma\sqrt{\log(N)/N}$ in Eqn. (2) for all arms $i \in [K]$, the regret of the HOPE algorithm is bounded as*

$$R(T) = O\left(T_0 + G_{\mathcal{S}_1,\hat{\boldsymbol{\theta}}}(T - T_0)\text{polylog}(T)/\sqrt{N}\right),$$

*where* polylog *denotes a polynomial term in the logarithm of the input, and $G_{\mathcal{S}_1,\hat{\boldsymbol{\theta}}}$ is a parameter that depends on the choice of the initial estimators $\{\hat{\boldsymbol{\theta}}^{(i)} : i \in [K]\}$ and the support estimations $\{\mathcal{S}_1^{(i)} : i \in [K]\}$, with its formal definition provided in App. D.2.*

The first term arises from exploration and the second from exploitation. This theorem is general in the sense that it is not restricted to any specific kind of sparsity, as in previous works. Moreover, it characterizes the performance under different choices of the initial estimator and the support estimation (as long as Assumps. 2 and 3 can be satisfied). Crucially, the proof relies on a new concentration bound for the PWE prediction error—not available in Zhao et al. (2023)—which we establish in Prop. 6. This yields a nonasymptotic guarantee for PWE in prediction settings.

To achieve further optimized performances, the choice of $N$ needs to be specified based on different scenarios. We then provide discussions under four scenarios: (1) with sparse model parameters; (2) with (approximately) sparse eigenvalues of $\boldsymbol{\Sigma}$; (3) with both kinds of sparsity; (4) with different kinds of sparsity for different arms. The first two *homogeneous* ones have been the focus of previous works, while the latter two are more challenging in their *heterogeneous* nature, which are studied for **the first time** by this work to the best of our knowledge.

## 6.1 SCENARIO 1: SPARSE MODEL PARAMETERS

First, in the parameter-sparse regime introduced in Sec. 2, HOPE attains regret guarantees that match the best-known results in the literature, thereby showing that our general framework subsumes the classical setting without loss in rate.

**Proposition 1** (Sparse Model Parameters)**.** *With Lasso as both the initial estimator and the support estimation, using $N \asymp K^{-2/3}s_0^{1/3}T^{2/3}$, under Assump. 1 and the conditions in App. G.1, G.2 for the guarantee of Lasso, the regret of HOPE is bounded as*

$$R(T) = O\left(K^{\frac{1}{3}}s_0^{\frac{1}{3}}T^{\frac{2}{3}}\text{polylog}(T)\right).$$

Prior work on high-dimensional sparse linear contextual bandits reports regret bounds under varying assumptions and settings (see App. B). The most directly comparable result is provided by Li et al. (2022), which achieves $O\left(K^{1/3}s_0^{1/3}T^{2/3}\text{polylog}(pT)\right)$.

**Remark 3.** *In this scenario, the exploration length $T_0 = NK$ is chosen with knowledge of $s_0$, a common assumption in high-dimensional sparse bandits (Bastani & Bayati, 2020; Hao et al., 2020; Li et al., 2022; Wang et al., 2018; Lee et al., 2024). Importantly, HOPE also admits* sparsity-agnostic *tuning. For Scenario 1, setting $N \asymp K^{-2/3}T^{2/3}$ (independent of $s_0$) yields a regret of order $\tilde{O}\left(K^{1/3}s_0^{1/2}T^{2/3}\right)$—incurring only a minor $s_0^{1/6}$ overhead relative to the $s_0^{1/3}$ rate—while preserving sublinear regret. Analogous agnostic choices apply to the other scenarios; see App. F for details.*

## 6.2 SCENARIO 2: (APPROXIMATELY) SPARSE EIGENVALUES OF CONTEXT COVARIANCE MATRICES

We next consider the scenario where the context covariance matrices have (approximately) sparse eigenvalues. For concreteness, the results of HOPE under Example 1(A) are stated below; the corresponding result for Example 1(B) is deferred to App. D.1.

**Proposition 2** (Sparse Eigenvalues of $\boldsymbol{\Sigma}$: Example 1(A)). *With RDL as the initial estimator and* $\mathcal{S}_1^{(i)} = [p]$ *for all arms, using* $N \asymp \max\{K^{-\frac{1}{2}} p^{\frac{1}{2T^a}} T^{\frac{a+2}{4}}, K^{-\frac{2}{3}} p^{\frac{2}{3T^a}} T^{\frac{2-a}{3}}\}$, *under Assump. 1 and the conditions in App. G.3 for the guarantee of RDL, if the covariance matrices satisfy Example 1(A), the regret of HOPE is bounded as*

$$R(T) = \tilde{O}\big( \max\big\{ K^{\frac{1}{2}} p^{\frac{1}{2T^a}} T^{\frac{a+2}{4}}, K^{\frac{1}{3}} p^{\frac{2}{3T^a}} T^{\frac{2-a}{3}} \big\}\big).$$

Under the same setting as Example 1(A), Komiyama & Imaizumi (2024) obtain a regret rate $\tilde{O}\big(K^{2/3} T^{\max\{(2+a)/3,\, 1-a/2\}}\big)$. In our bound, the factors $p^{1/(2T^a)}$ and $p^{2/(3T^a)}$ are subpolynomial in $p$; indeed, for fixed $a > 0$ and any constant $c > 0$, $p^{c/T^a} \to 1$ as $T \to \infty$. Under the mild growth condition $\log p \leq \frac{1}{2} T^a \log T$, we further have $p^{c/T^a} \leq T^{c/2}$, so the regret of HOPE is effectively polynomial only in $K$ and $T$, and improves on the above rate. The comparison under Example 1(B) is analogous and can be found in App. D.1.

## 6.3 SCENARIO 3: BOTH SPARSITIES

We consider a scenario absent from prior work in which *both* sources of structure are present: the model parameters $\{\boldsymbol{\theta}^{(i)}\}_{i=1}^K$ are sparse and the context covariances $\{\boldsymbol{\Sigma}^{(i)}\}_{i=1}^K$ have (approximately) sparse eigenvalues. For any positive semidefinite (PSD) matrix $A$, define $\mathrm{M}(\mathbf{A}) := \mathrm{tr}(\mathbf{A})/\|\mathbf{A}\|_F$, so that $\mathrm{M}(\mathbf{A})^2 = \mathrm{erank}(\mathbf{A})$ (effective rank). Let $M_i := \mathrm{M}(\boldsymbol{\Sigma}_i[\mathcal{S}_1^{(i)}])$ and $M := \max_{i \in [K]} M_i$. Intuitively, $M$ measures spectral complexity on the learned support—small when only a few eigenvalues carry most of the mass.

**Proposition 3** (Both Sparsities). *With Lasso as the initial estimator and also Lasso to perform the support estimation, using* $N \asymp K^{-2/3} M^{2/3} T^{2/3}$, *under Assump. 1 and the conditions in App. G.1, G.2 for the guarantees of Lasso, if the eigenvalues of covariance matrices* $\boldsymbol{\Sigma}^{(i)}$ *for all* $i \in [K]$ *decay sufficiently fast (e.g., Example 1; see App. D.6 for details), the regret of HOPE is bounded as*

$$R(T) = \tilde{O}\big(K^{\frac{1}{3}} M^{\frac{2}{3}} T^{\frac{2}{3}}\big).$$

**Remark 4** (Comparison with Scenario 1). *Since* $M_i^2 = \mathrm{erank}(\boldsymbol{\Sigma}_i[\mathcal{S}_1^{(i)}]) \leq \mathrm{rank}(\boldsymbol{\Sigma}_i[\mathcal{S}_1^{(i)}]) \leq |\mathcal{S}_1^{(i)}|$, *we have* $M \leq \max_i \sqrt{|\mathcal{S}_1^{(i)}|}$. *By the Lasso support-size control (App. G.2),* $|\mathcal{S}_1^{(i)}| \lesssim s_0$ *with high probability (w.h.p.), hence* $M \lesssim \sqrt{s_0}$ *w.h.p.; with eigenvalue decay,* $M$ *is typically much smaller than* $\sqrt{s_0}$. *Consequently, Prop. 3 improves upon the parameter-only rate* $\tilde{O}(K^{1/3} s_0^{1/3} T^{2/3})$ *(cf. Prop. 1 and Li et al., 2022) whenever* $M < \sqrt{s_0}$—*capturing the gain from jointly exploiting parameter and spectral sparsity.*

## 6.4 SCENARIO 4: MIXED SPARSITIES

Finally, we consider a mixed-sparsity setting: a subset of arms (Part I) has sparse parameters $\{\boldsymbol{\theta}^{(i)}\}$ with sparsity level $s_0$, whereas the remaining arms (Part II) exhibit (approximately) sparse eigenvalues in their context covariances $\{\boldsymbol{\Sigma}^{(i)}\}$. In contrast to prior work, sparsity types vary across arms.

**Proposition 4** (Mixed sparsity). *Consider HOPE configured as follows: Lasso serves as both the initial estimator and the support selector for Part I; RDL serves as the initial estimator for Part II with* $\mathcal{S}_1^{(i)} = [p]$. *With N chosen as in Eqn.* (16), *under Assump. 1, the Lasso guarantees in Apps. G.1–G.2 (Part I), and the RDL guarantee in App. G.3 (Part II), if the Part II covariances satisfy Example 1(A), then*

$$R(T) = \tilde{O}\Big(\max\Big\{ K^{\frac{1}{3}} s_0^{\frac{1}{3}} T^{\frac{2}{3}},\ K^{\frac{1}{2}} p^{\frac{1}{2T^\alpha}} T^{\frac{a+2}{4}},\ K^{\frac{1}{3}} p^{\frac{2}{3T^\alpha}} T^{\frac{2-a}{3}} \Big\}\Big).$$

*An analogous bound holds when Part II satisfies Example 1(B).*

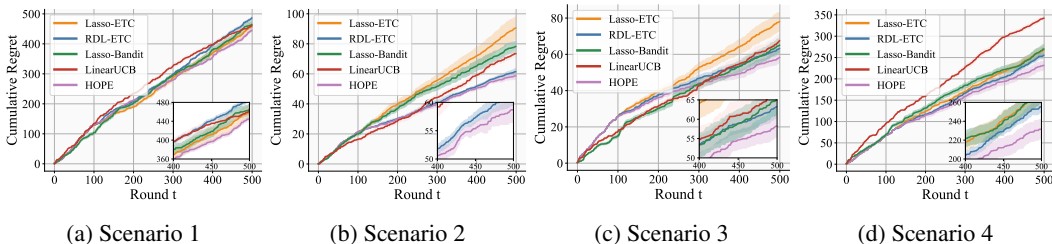

| (a) Scenario 1 | (b) Scenario 2 | (c) Scenario 3 | (d) Scenario 4 |

Figure 2: Comparison of methods on four different cases. A smaller regret indicates better performance. The solid lines are the mean of 10 repetitions, and the bands represent the standard deviation.

It can be observed that the performance of HOPE is dominated by the worst among the two groups of arms under this scenario. Moreover, we note that ***none*** of the previous works can handle this scenario as their approaches are confined to only one type of sparsity (Li et al., 2022; Komiyama & Imaizumi, 2024; Bastani & Bayati, 2020; Wang et al., 2018).

# 7 EXPERIMENTS

## 7.1 EXPERIMENT SETTINGS

We compare HOPE with Lasso-ETC (Li et al., 2022), RDL-ETC (Komiyama & Imaizumi, 2024), Lasso-Bandit (Bastani & Bayati, 2020), and LinearUCB (Chu et al., 2011) under four settings. For all experiments, we set $K = 5, T = 500$, and $p = 200$. We denote the sparsity ratio $r(\boldsymbol{\theta})$ of $\boldsymbol{\theta}$ as $s_0/p$. The non-zero elements of all arms are sampled from a standard normal distribution. The covariance matrix $\boldsymbol{\Sigma}^{(i)}$ and $r(\boldsymbol{\theta}^{(i)})$ for arm $i$ are configured as follows: ❶ **Scenario 1**(§ 6.1): We set $\boldsymbol{\Sigma}^{(i)} = \mathbf{I}$ and $r(\boldsymbol{\theta}^{(i)}) = 0.1$ for $i \in [K]$. ❷ **Scenario 2**(§ 6.2): We set $\boldsymbol{\Sigma}^{(i)} = c^{(i)}\mathbf{diag}(\lambda_1, .., \lambda_p)$ with $c^{(i)} \sim \mathrm{Uni}[0.5, 1.5]$, where $\lambda_k = k^{-1+\frac{1}{T}}$. We set $r(\boldsymbol{\theta}^{(i)}) = 0.9$ for $i \in [K]$. ❸ **Scenario 3**(§ 6.3): We set $\boldsymbol{\Sigma}^{(i)} = c^{(i)}\boldsymbol{\Sigma}$ with $c^{(i)} \sim \mathrm{Uni}[0.5, 1.5]$ and $\boldsymbol{\Sigma} = \mathbf{diag}(\lambda_1, .., \lambda_p)$, where $\lambda_k = k^{-1+\frac{1}{T}}$, but we set $r(\boldsymbol{\theta}^{(i)}) = 0.1$ for $i \in [K]$. ❹ **Scenario 4**(§ 6.4): We set $r(\boldsymbol{\theta}_1) = r(\boldsymbol{\theta}_2) = 0.1$ and $\boldsymbol{\Sigma}^{(1)} = \boldsymbol{\Sigma}^{(2)} = \mathbf{I}$. For the remaining three arms, we set $r(\boldsymbol{\theta}^{(i)}) = 0.9$ with $\boldsymbol{\Sigma}^{(i)} = c^{(i)}\mathrm{diag}(\lambda_1, .., \lambda_p)$ and $\lambda_k = k^{-1+\frac{1}{T}}$. We generate $\mathbf{X}^{(i)}(t)$ from $N(0, \boldsymbol{\Sigma}^{(i)})$ and compute $\boldsymbol{y}^{(i)}(t) = \mathbf{X}^{(i)}(t)\boldsymbol{\theta}^{(i)} + \boldsymbol{\varepsilon}$ with the noise $\boldsymbol{\varepsilon} \sim N(0, 0.1\mathbf{I})$.

## 7.2 EXPERIMENT RESULTS

Fig. 2 shows the results of our proposed HOPE algorithm alongside other high-dimensional ETC algorithms in four scenarios. Our key observations are: **(1) Comparable Performance in Homogeneous Scenarios:** HOPE matches the performance of existing algorithms in Scenarios 1 and 2, which are well-studied. By leveraging the initial estimator, HOPE selects the most suitable method for final predictions. **(2) Superior Performance in Heterogeneous Scenarios:** In Scenario 3, where both model parameters and eigenvalues of $\boldsymbol{\Sigma}^{(i)}$ exhibit sparsity, HOPE outperforms Lasso-ETC and RDL-ETC by utilizing both sparsity types. In Scenario 4, varying sparsity ratios challenge other methods; for instance, Lasso-ETC struggles to adapt to non-sparse scenarios. HOPE consistently excels due to the adaptability of the PWE approach.

# 8 CONCLUSIONS

Existing high-dimensional linear contextual bandit algorithms typically focus on one specific structure, either sparse model parameters or (appr.) sparse eigenvalues of the context covariance matrices. In this work, we integrate the PWE estimator, which adapts to both types of sparsity, into the bandit framework and, building on this, propose HOPE. Comprehensive theoretical analyses highlight the effectiveness and flexibility of HOPE. In two existing homogeneous scenarios, HOPE achieved comparable results to those of previous approaches. In two newly proposed challenging heterogeneous scenarios, HOPE can still perform in a theoretically efficient manner while previous approaches fail. Empirical studies further demonstrated the superiority and adaptability of HOPE across various scenarios. To the best of our knowledge, HOPE is the first to effectively address both types of sparsity in high-dimensional contextual bandit problems.

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

## A  RELATED WORKS

**High-dimensional linear contextual bandits.**  To address the curse of dimensionality, research in this field often incorporates additional structural assumptions Wang et al. (2018); Kim & Paik (2019); Bastani & Bayati (2020); Deshpande & Montanari (2012); Chen et al. (2021); Hamidi et al. (2019); Shi et al. (2021). One prevalent assumption is that the model parameters exhibit sparsity. Various tools are employed, including Lasso regression Bastani & Bayati (2020); Ren & Zhou (2024); Hao et al. (2020); Oh et al. (2021); Li et al. (2022), subset selection methods Wang et al. (2020), and Thompson sampling techniques Chakraborty et al. (2023). In contrast, Komiyama & Imaizumi (2024) study the sparse structure of context covariance eigenvalues and propose an algorithm based on RDL Bartlett et al. (2020), achieving sublinear regret rates in low-rank scenarios. While these approaches demonstrate effectiveness in homogeneous settings where all arms have one same type of sparsity, they face limitations in more heterogeneous contexts, e.g., arms have two types of sparsity at the same time (as in Sec. 6.3) or different arms have different types of sparsity (as in Sec. 6.4). This variability restricts their flexibility and applicability.

**High-dimensional linear regression.**  Various regularization techniques for sparsity settings, such as Lasso and other penalized methods, have been proposed Tibshirani (1996); Zou & Hastie (2005); Fan & Li (2001); Zhang (2010). Theoretical foundations for these scenarios are well-established in the literature Wainwright (2019); Vershynin (2018); Zhang (2023). Beyond that, researchers begin exploring overparameterized settings using the ridgeless ordinary least squares (OLS) estimator, which employs the Moore-Penrose generalized inverse to effectively handle non-sparse scenarios Bartlett et al. (2020); Azriel & Schwartzman (2020); Hastie et al. (2022). However, these methods often estimate high-dimensional parameters directly with insufficient data, resulting in suboptimal performances. In contrast, Zhao et al. (2023) focus on the final reward as an unknown parameter for prediction. This approach reduces the model to fewer parameters, making it more tractable and solvable.

## B  COMPARISON OF HOPE WITH EXISTING WORKS

To clarify the comparison, we first formulate the setting of our work and then outline the settings of previous works Hao et al. (2020); Li et al. (2022); Bastani & Bayati (2020), highlighting how they relate to our models.

**This Work and Komiyama & Imaizumi (2024): Finite Heterogeneous Arms, Stochastic Heterogeneous Contexts.**  We follow the problem formulation in Komiyama & Imaizumi (2024). Specifically, We consider $K$ $p$-dimensional parameters $\{\boldsymbol{\theta}_i\}_{i=1}^K$, one for each arm (thus referred to as "finite heterogeneous arms"). At each time $t \in [T]$, a set of $K$ $p$-dimensional contexts $\{\boldsymbol{x}_{t,i}\}_{i=1}^K$ is generated, also one for each arm (thus referred to as "stochastic heterogeneous contexts"). The agent then selects an action $a_t \in [K]$ and receives a reward:

$$y_t = \langle \boldsymbol{\theta}_{a_t}, \boldsymbol{x}_{t,a_t} \rangle + \varepsilon_t.$$

**Li et al. (2022); Lee et al. (2024): Finite Homogeneous Arms, Stochastic Heterogeneous Contexts.**  One model parameter $\boldsymbol{\beta} \in \mathbb{R}^{p'}$ is considered, which is shared among all arms (thus referred to as "finite homogeneous arms"). At each time $t \in [T]$, a set of $K$ $p'$-dimensional contexts $\{\boldsymbol{z}_{t,i}\}_{i=1}^K$ is generated, one for each arm. The agent then selects an action $a_t \in [K]$ and receives a reward:

$$y_t = \langle \boldsymbol{\beta}, \boldsymbol{z}_{t,a_t} \rangle + \varepsilon_t.$$

This setting, due to its homogeneity, can be understood as a degeneration of the one considered in this work (i.e., restricting $\boldsymbol{\theta}_i = \boldsymbol{\beta}, \forall i \in [K]$). From another perspective, it can be translated into the heterogeneous setting by consider $p' = Kp$, $\boldsymbol{\beta} = [\boldsymbol{\theta}_1^\top, \cdots, \boldsymbol{\theta}_K^\top]^\top$, and $\boldsymbol{z}_{t,i} = [\boldsymbol{0}^\top, \ldots, \boldsymbol{0}^\top, \boldsymbol{x}_{t,i}^\top, \boldsymbol{0}^\top, \ldots, \boldsymbol{0}^\top]^\top$ (i.e., with $\boldsymbol{x}_{t,i}^\top$ occupying positions in $[(i-1)p+1, ip]$).

In the sparse scenario, the regret bound obtained in Li et al. (2022) is $O(s_0^{1/3} T^{2/3} \text{polylog}(p'T))$. After converting the settings with the above transformation, their regret bound becomes $O(K^{\frac{1}{3}} s_0^{\frac{1}{3}} T^{\frac{2}{3}} \text{polylog}(KpT))$, worse than our bound $O(K^{\frac{1}{3}} s_0^{\frac{1}{3}} T^{\frac{2}{3}} \text{polylog}(T))$ in Proposition 1 in

the high-dimensional scenario with $T \ll p$. Moreover, when the eigenvalues of the covariance matrix decay rapidly, our regret bound improves to $\tilde{O}(K^{\frac{1}{2}} s_0^{\frac{1}{2}} T^{\frac{1}{2}})$, offering a significant advantage. These results demonstrate that our approach not only achieves a comparable regret bound but, in some cases, provides superior performance, underscoring its effectiveness in the high-dimensional contextual bandit setting.

The regret of Lee et al. (2024) is $O(s_0^2 \log(p'T) \log T)$. After converting the settings with the above transformation, their regret bound becomes $O(K^2 s_0^2 \log(Kp'T) \log T)$. While their bound demonstrates better dependence on the time horizon $T$, it exhibits worse scaling with respect to both $K$ and $s_0$ compared to our results. Moreover, their theoretical guarantees **require an additional margin condition** (i.e., Assumption 2 in Lee et al. (2024)), which imposes stricter requirements on the problem structure than our framework. This assumption is unnecessary for our theoretical analysis. Due to these fundamental differences in problem setup and theoretical requirements, a direct comparison between the two results would be inappropriate.

**Bastani & Bayati (2020); Wang et al. (2018): Finite Heterogeneous Arms, Stochastic Homogeneous Contexts.** Given $K$ $p''$-dimensional vectors $\{\boldsymbol{\beta}_i\}_{i=1}^K$ (i.e., finite heterogeneous arms), the model generates a $p''$-dimensional context $\boldsymbol{z}_t$ at each time $t \in [T]$, which is shared among all arms (thus referred to as "stochastic homogeneous contexts"). The agent chooses an action $a_t \in [K]$ and receives a reward:

$$y_t = \langle \boldsymbol{\beta}_{a_t}, \boldsymbol{z}_t \rangle + \varepsilon_t.$$

Similarly as abovementioned, this setting can also be viewed as a degenerated one from the setting in this work (i.e., restricting $\boldsymbol{x}_{t,i} = \boldsymbol{z}_t, \boldsymbol{\theta}_i = \boldsymbol{\beta}_i, \forall i \in [K]$. Also, it can be translated into the setting in this work by considering $p'' = Kp$, $\boldsymbol{\beta}_i = [\boldsymbol{0}^\top, \ldots, \boldsymbol{0}^\top, \boldsymbol{\theta}_i^\top, \boldsymbol{0}^\top, \ldots, \boldsymbol{0}^\top]^\top$ and $\boldsymbol{z}_t = [\boldsymbol{x}_{t,1}^\top, \ldots, \boldsymbol{x}_{t,K}^\top]^\top$.

It is noted that the regret bound obtained in Bastani & Bayati (2020) only has logarithmic dependency on $T$, instead of the polynomial ones in this work $\mathcal{O}(\tau K s^2 \log^2 T)$; however, Bastani & Bayati (2020) requires additional margin and constant gap conditions for competitive arms, which are stricter than the assumptions in our setting and not required for our theory. Due to such unfairness, the results are non-comparable.

**Hao et al. (2020): (Potentially) Infinite Homogeneous Arms, Fixed Heterogeneous Contexts** The model considers a shared model parameter $\boldsymbol{\beta} \in \mathbb{R}^{p^\dagger}$ shared among all arms, and a compact action set $\mathcal{Z} \subset \mathbb{R}^{p^\dagger}$ (which is fixed in all time steps). At each time $t$, the agent selects an action $\boldsymbol{z}_t \in \mathcal{Z}$ and receives a reward:

$$y_t = \langle \boldsymbol{\beta}, \boldsymbol{z}_t \rangle + \varepsilon_t.$$

Our setting and theirs, in general, can**not** be converted into each other due to the different considerations of arms and contexts. In particular, their analysis fundamentally relies on the assumption of fixed arm contexts, whereas our approach accommodates stochastic arms contexts.

## C OMITTED ALGORITHMIC DETAILS

In the main paper, there are two components introduced in the design of PWE, i.e., Algorithm 1, without discussions: the estimated support set $\mathcal{S}_1^{(i)}$ and the bases $\boldsymbol{\Gamma}_t^{(i)}$ for sparsification, which are further illustrated in the following.

### C.1 ESTIMATING THE SUPPORT SET

#### C.1.1 INTUITIONS

The following observation motivates us to consider estimators using the information of the sparsity degree of $\boldsymbol{\theta}^{(i)}$. For any subset $\mathcal{S}_1^{(i)} \subseteq \{1, \ldots, p\}$ such that $\mathcal{S}_0^{(i)} \subseteq \mathcal{S}_1^{(i)}$, we observe that

$$\mu_t^{(i)} := \langle \boldsymbol{\theta}^{(i)}, \boldsymbol{x}_t^{(i)} \rangle = \langle \boldsymbol{\theta}^{(i)}[\mathcal{S}_0^{(i)}], \boldsymbol{x}_t^{(i)}[\mathcal{S}_0^{(i)}] \rangle = \langle \boldsymbol{\theta}^{(i)}[\mathcal{S}_1^{(i)}], \boldsymbol{x}_t^{(i)}[\mathcal{S}_1^{(i)}] \rangle = \langle \boldsymbol{\theta}^{(i)}, \tilde{\boldsymbol{x}}_{t,\mathcal{S}_1^{(i)}}^{(i)} \rangle =: \mu_{t,\mathcal{S}_1^{(i)}}^{(i)},$$

where $\boldsymbol{x}_t^{(i)}[\mathcal{S}_0^{(i)}]$ and $\boldsymbol{\theta}^{(i)}[\mathcal{S}_0^{(i)}]$ (similarly, $\boldsymbol{x}_t^{(i)}[\mathcal{S}_1^{(i)}]$ and $\boldsymbol{\theta}^{(i)}[\mathcal{S}_1^{(i)}]$) are the sub-vectors $\boldsymbol{x}_t^{(i)}$ and $\boldsymbol{\theta}^{(i)}$ truncated with elements contained in $\mathcal{S}_0$ (similarly, $\mathcal{S}_1$), respectively, and $\tilde{\boldsymbol{x}}_{t,\mathcal{S}_1^{(i)}}^{(i)}$ is a $p$-dimensional vector obtained by setting the elements of $\boldsymbol{x}_t^{(i)}$ that are not in $\mathcal{S}_1$ to be zero. Thus, instead of estimating $\mu_t^{(i)}$, we can equivalently consider the prediction at the point $\tilde{\boldsymbol{x}}_{t,\mathcal{S}_1^{(i)}}^{(i)}$, which is a sparse vector when $s_1^{(i)} = |\mathcal{S}_1^{(i)}|$ is small.

### C.1.2 TWO ESTIMATION TECHNIQUES

Then, we introduce two techniques for selecting the support set $\mathcal{S}_1^{(i)}$, which is used in Algorithm 1. Specifically, we explain how Lasso (Tibshirani, 1996) and Sure Independence Screening (SIS) (Fan & Lv, 2008) are applied in the context of our model.

**Lasso.** Lasso (Least Absolute Shrinkage and Selection Operator) (Tibshirani, 1996) provides another approach for variable selection, which simultaneously performs regression and selection by adding an $l_1$-norm regularization term to the least squares loss function. In our setting, given the arm contexts $\mathbf{X}^{(i)} \in \mathbb{R}^{N \times p}$ and the rewards $\boldsymbol{y}^{(i)} \in \mathbb{R}^N$, Lasso solves the following optimization problem for arm $i$ as discussed in the main paper:

$$\hat{\boldsymbol{\theta}}_{\text{Lasso}}^{(i)} = \arg\min_{\boldsymbol{\theta} \in \mathbb{R}^p} \left\{ \|\boldsymbol{y}^{(i)} - \mathbf{X}^{(i)}\boldsymbol{\theta}\|_2^2 + \lambda\|\boldsymbol{\theta}\|_1 \right\}.$$

where $\lambda > 0$ is a regularization parameter. After solving the Lasso optimization, the selected set $\mathcal{S}_1^{(i)}$ consists of the indices corresponding to the non-zero entries in $\hat{\boldsymbol{\theta}}^{(i)} = [\hat{\theta}_1^{(i)}, \cdots, \hat{\theta}_p^{(i)}]$, i.e.,

$$\mathcal{S}_1^{(i)} = \left\{ k \in [p] : \hat{\theta}_k^{(i)} \neq 0 \right\}.$$

**SIS** SIS (Sure Independence Screening) (Fan & Lv, 2008) is a two-step procedure designed for high-dimensional data, particularly when $p \gg N$. In our setup, where $\mathbf{X}^{(i)} \in \mathbb{R}^{N \times p}$ represents the collected arm contexts for arm $i$, SIS computes the marginal correlation between each predictor $\mathbf{X}_{:,k}^{(i)}$ and the reward vector $\boldsymbol{y}^{(i)}$. The marginal correlation is defined as:

$$\hat{\rho}_k^{(i)} = \frac{1}{N} \sum_{\tau=1}^{N} x_{\tau,k}^{(i)} y_\tau^{(i)} \text{ for } k \in [p],$$

where $x_{\tau,k}^{(i)}$ is the $k$-th predictor (i.e., the context feature at $k$-th dimension) for arm $i$ at time $\tau$, and $y_\tau^{(i)}$ is the corresponding reward.

SIS selects a subset of predictors $\mathcal{S}_1^{(i)} \subseteq [p]$ by ranking the predictors based on the magnitude of their marginal correlations as:

$$\mathcal{S}_1^{(i)} = \left\{ k \in [p] : |\hat{\rho}_k^{(i)}| \geq \tau_{\text{SIS}} \right\},$$

where $\tau_{\text{SIS}}$ is a threshold chosen to ensure that the size of the selected set is small, typically $|\mathcal{S}_1^{(i)}| = s_1^{(i)} \ll p$.

**Remark 5.** Additionally, we note that a two-step procedure can be employed: first, applying SIS to quickly reduce the dimensionality of the problem, and then using Lasso to further refine the selection of important predictors. This combined approach is highly efficient in high-dimensional settings.

### C.2 CONSTRUCTING BASIS FOR SPARISIFICATION

In this section, we discuss the detailed construction of $\boldsymbol{\Gamma}_t^{(i)}$. To facilitate discussions, the notation $\boldsymbol{\lambda}(\mathbf{A})$ is introduced to denote the vector of eigenvalues of a positive semi-definite matrix $\mathbf{A} \in \mathbb{R}^{m \times m}$, arranged in decreasing order. We consider $\boldsymbol{\lambda}(\mathbf{A})$ to be approximately sparse when only a few eigenvalues are significantly larger than $m^{-1} \sum_j \boldsymbol{\lambda}_j(\mathbf{A})$. Recall that

$$\boldsymbol{\xi}_t^{(i)} = (\boldsymbol{\Gamma}_t^{(i)})^{-1} \boldsymbol{\zeta}_t^{(i)} = (\boldsymbol{\Gamma}_t^{(i)})^{-1} \frac{\mathbf{X}^{(i)} \mathbf{Q}_t^{(i)} \boldsymbol{\theta}^{(i)}}{\sqrt{N}}.$$

Since $\boldsymbol{\zeta}_t^{(i)}$ is in general a non-sparse vector, we aim to properly choose $\boldsymbol{\Gamma}_t^{(i)}$ such that $\boldsymbol{\xi}_t^{(i)}$ is (approximately) sparse. Specially, we focus on two different sources of information: the approximate sparse eigenvalues of the covariance matrix $\boldsymbol{\Sigma}^{(i)}$ and the potential sparsity of the initial parameter vector $\boldsymbol{\theta}^{(i)}$.

For the first information, i.e., the approximate sparse eigenvalues of the covariance matrix $\boldsymbol{\Sigma}^{(i)}$, given the projection matrix $\mathbf{Q}_t^{(i)}$, we have the following relationship:

$$\boldsymbol{\lambda}_j\left(N^{-1}\mathbf{X}^{(i)}\mathbf{Q}_t^{(i)}(\mathbf{X}^{(i)})^\top\right) = \boldsymbol{\lambda}_j\left(N^{-1}\mathbf{X}^{(i)}\mathbf{Q}_t^{(i)}(\mathbf{Q}_t^{(i)})^\top(\mathbf{X}^{(i)})^\top\right)$$
$$= \boldsymbol{\lambda}_j\left(N^{-1}\mathbf{Q}_t^{(i)}(\mathbf{X}^{(i)})^\top\mathbf{X}^{(i)}\mathbf{Q}_t^{(i)}\right),\ \forall j \in [N].$$

Note that the population version of $\boldsymbol{\lambda}\left(N^{-1}\mathbf{Q}_t^{(i)}(\mathbf{X}^{(i)})^\top\mathbf{X}^{(i)}\mathbf{Q}_t^{(i)}\right)$ is $\boldsymbol{\lambda}\left(\mathbf{Q}_t^{(i)}\boldsymbol{\Sigma}^{(i)}\mathbf{Q}_t^{(i)}\right)$. Let $\boldsymbol{\Gamma}_{\mathrm{eg}}\boldsymbol{\Psi}\boldsymbol{\Gamma}_{\mathrm{eg}}^\top$ denote the spectral decomposition of $N^{-1}\mathbf{X}^{(i)}\mathbf{Q}_t^{(i)}(\mathbf{X}^{(i)})^\top$ with $\boldsymbol{\Gamma}_{\mathrm{eg}} = [\boldsymbol{u}_{\mathrm{eg},1}, \ldots, \boldsymbol{u}_{\mathrm{eg},N}] \in \mathbb{R}^{N\times N}$ representing the eigenvectors, and $\boldsymbol{\Psi} = \mathrm{diag}\left(\psi_1, \ldots, \psi_N\right)$ containing the corresponding eigenvalues in decreasing order. Then $\boldsymbol{u}_{\mathrm{eg},i}, i \in [N]$ are also the left-singular vectors of $\mathbf{X}^{(i)}\mathbf{Q}_t^{(i)}$. For $\boldsymbol{\Gamma}_t^{(i)} = \boldsymbol{\Gamma}_{\mathrm{eg}}$, the non-zero elements of $\boldsymbol{\xi}_t^{(i)}$ would concentrate on the significant eigenvalues of $\boldsymbol{\lambda}\left(\mathbf{Q}_t^{(i)}\boldsymbol{\Sigma}^{(i)}\mathbf{Q}_t^{(i)}\right)$. If $\boldsymbol{\lambda}(N^{-1}\mathbf{X}^{(i)}\mathbf{Q}_t^{(i)}\mathbf{X}^{(i)\top})$ is approximately sparse (i.e., $\boldsymbol{\lambda}\left(\mathbf{Q}_t^{(i)}\boldsymbol{\Sigma}^{(i)}\mathbf{Q}_t^{(i)}\right)$ is approximately sparse), $\boldsymbol{\xi}_t^{(i)}$ will also be approximately sparse regardless $\boldsymbol{\zeta}_t^{(i)}$ being sparse or not (Zhao et al., 2023).

For the second information, i.e., the potential sparsity of the initial parameter vector $\boldsymbol{\theta}^{(i)}$, if a reliable initial estimator $\hat{\boldsymbol{\theta}}^{(i)}$ is available, e.g., a Lasso estimator in sparse model parameter settings, we can further use

$$\boldsymbol{\zeta}_{\hat{\boldsymbol{\theta}}^{(i)}} = N^{-1/2}\mathbf{X}^{(i)}\boldsymbol{Q}_t^{(i)}\hat{\boldsymbol{\theta}}^{(i)}$$

as an estimate of $\boldsymbol{\zeta}_t^{(i)}$.

To leverage both sources of information jointly, we construct $\boldsymbol{\Gamma}_t^{(i)}$ by replacing one of the columns (e.g., the $m$-th column) of $\boldsymbol{\Gamma}_{\mathrm{eg}}$ with $\bar{\boldsymbol{\zeta}}_{\hat{\boldsymbol{\theta}}^{(i)}} = \boldsymbol{\zeta}_{\hat{\boldsymbol{\theta}}^{(i)}}/\|\boldsymbol{\zeta}_{\hat{\boldsymbol{\theta}}^{(i)}}\|_2$,

$$\boldsymbol{\Gamma}_t^{(i)}(\hat{\boldsymbol{\theta}}^{(i)}) = \left[\boldsymbol{u}_{\mathrm{eg},1}, \cdots, \boldsymbol{u}_{\mathrm{eg},m-1}, \bar{\boldsymbol{\zeta}}_{\hat{\boldsymbol{\theta}}^{(i)}}, \boldsymbol{u}_{\mathrm{eg},m+1}, \cdots, \boldsymbol{u}_{\mathrm{eg},N}\right],$$

which is an empirical counterpart of

$$\boldsymbol{\Gamma}_t^{(i)}(\boldsymbol{\theta}^{(i)}) = \left[\boldsymbol{u}_{\mathrm{eg},1}, \cdots, \boldsymbol{u}_{\mathrm{eg},m-1}, \bar{\boldsymbol{\zeta}}_{\boldsymbol{\theta}^{(i)}}, \boldsymbol{u}_{\mathrm{eg},m+1}, \cdots, \boldsymbol{u}_{\mathrm{eg},N}\right],$$

To mitigate the collinearity between $\boldsymbol{z}_t^{(i)}$ and other predictors in the transformed model, we replace $\boldsymbol{u}_{\mathrm{eg},i_0}$ with $\bar{\boldsymbol{\zeta}}_{\hat{\boldsymbol{\theta}}^{(i)}}$, where $i_0 = \arg\max_{1\le i\le N}\left|\boldsymbol{u}_{\mathrm{eg},i}^\top\boldsymbol{z}_t^{(i)}\right|$. The non-singular property of $\boldsymbol{\Gamma}_t^{(i)}(\hat{\boldsymbol{\theta}}^{(i)})$ is discussed in Zhao et al. (2023).

The sparsity of $\boldsymbol{\xi}_t^{(i)}$ is influenced by both $\hat{\boldsymbol{\theta}}^{(i)}$ and the sparsity of $\boldsymbol{\lambda}(N^{-1}\mathbf{X}^{(i)}\mathbf{Q}_t^{(i)}\mathbf{X}^{(i)\top})$. In the ideal case where $\hat{\boldsymbol{\theta}}^{(i)} = \boldsymbol{\theta}^{(i)}$, it can be shown that $\boldsymbol{\xi}_t^{(i)} := \boldsymbol{\Gamma}_t^{(i)}(\hat{\boldsymbol{\theta}}^{(i)})^{-1}\boldsymbol{\zeta}_{\boldsymbol{\theta}^{(i)}} \propto (1,0,...,0)^\top$, resulting in a sparse vector. That means if we can well estimate $\boldsymbol{\theta}^{(i)}$, then $(\boldsymbol{u}_{\mathrm{eg},j}, j \neq= i_0)$ do not help much. However, when $\hat{\boldsymbol{\theta}}^{(i)}$ is not good enough (e.g., $\boldsymbol{\theta}^{(i)}$ is not sufficiently sparse) but $\boldsymbol{\lambda}(N^{-1}\mathbf{X}^{(i)}\mathbf{Q}_t^{(i)}\mathbf{X}^{(i)\top})$ is sufficiently sparse, the inclusion of $\boldsymbol{u}_{\mathrm{eg}}$'s will compensate the inaccuracies of $\hat{\boldsymbol{\theta}}^{(i)}$. Thus, both sources of information can enhance each other, making the estimator more robust to underlying assumptions.

For HOPE, i.e., Algorithm 2, we consider two initial estimators $\hat{\boldsymbol{\theta}} \in \{\hat{\boldsymbol{\theta}}_{\mathrm{lasso}}, \hat{\boldsymbol{\theta}}_{\mathrm{rdl}}\}$ to construct $\boldsymbol{\Gamma}_t^{(i)}(\hat{\boldsymbol{\theta}})$. Especially, a standard cross-validation procedure can be performed to select a more accurate estimator. Zhao et al. (2023) also incorporates two additional choices for $\boldsymbol{\Gamma}_t^{(i)}$, i.e., $\boldsymbol{\Gamma}_t^{(i)}(\hat{\boldsymbol{\theta}}_{\mathrm{ridge}})$ and $\boldsymbol{\Gamma}_{\mathrm{eg}}$, both of which can also be applied. For further details on these two choices, please refer to Zhao et al. (2023)

## D  THEORETICAL RESULTS AND PROOFS

### D.1  ADDITIONAL RESULTS FOR (APPROXIMATELY) SPARSE EIGENVALUES SCENARIO

**Proposition 5** (Sparse Eigenvalues of $\boldsymbol{\Sigma}$: Example 1(B)). *With RDL as the initial estimator and* $\mathcal{S}_1^{(i)} = [p]$ *for all arms, using* $N \asymp \min(K^{-\frac{1}{2}}T^{\frac{1}{2}+\frac{c(2-b)}{4}}, K^{-\frac{1}{2}}T^{\frac{1}{2}+\frac{3c(1-b)}{4}})$, *under Assump. 1 and the additional conditions specified in App. G.3 for the guarantee of RDL, if the covariance matrices satisfy Example 1(B), the regret of HOPE is bounded as*

$$R(T) = \tilde{O}\big( \min \big\{ K^{\frac{1}{2}}T^{\frac{1}{2}+\frac{c(2-b)}{4}}, K^{\frac{1}{2}}T^{\frac{1}{2}+\frac{3c(1-b)}{4}} \big\}\big).$$

Again, under the same Example 1(B), the approach in (Komiyama & Imaizumi, 2024) obtains a regret of order $\tilde{O}(K^{\frac{2}{3}}T^{\frac{2+c(1-b)}{3}})$. It can be observed that the regret of HOPE is better when $b < 1/2$.

### D.2  NOTATIONS AND DEFINITIONS

In this section, we define the key parameter $G_{\mathcal{S}_1,\hat{\boldsymbol{\theta}}}$ in Theorem 1. Some notations are first introduced in the following. Let $\mathbf{A} \in \mathbb{R}^{m \times m}$ be a symmetric positive semidefinite matrix, with eigenvalues $\lambda_1(\mathbf{A}) \geq \cdots \geq \lambda_m(\mathbf{A})$. The smallest nonzero eigenvalue is denoted by $\lambda_{\min}^+(\mathbf{A})$.

**Definition 1** (Prediction Error). *For one estimator* $\hat{\boldsymbol{\theta}}^{(i)} \in \mathbb{R}^p$, *its prediction error with respect to* $\boldsymbol{\theta}^{(i)}$ *is defined as:*

$$d(\hat{\boldsymbol{\theta}}^{(i)}, \boldsymbol{\theta}^{(i)}) := \left( \mathrm{var}\left[ (\hat{\boldsymbol{\theta}}^{(i)} - \boldsymbol{\theta}^{(i)})^\top \boldsymbol{x}_t^{(i)} \right] \right)^{1/2}.$$

**Definition 2.** *For a positive semi-definite matrix* $\mathbf{A} \in \mathbb{R}^{n \times n}$ *with positive eigenvalues* $\lambda_1(\mathbf{A}) \geq \cdots \geq \lambda_n(\mathbf{A}) \geq 0$, *we define*

$$\tilde{\lambda}_k(\mathbf{A}) := \frac{1}{n-k-1} \sum_{m=k+1}^n \lambda_m(\mathbf{A}),\ 0 \leq k \leq n-2,$$

$$\tilde{\lambda}_{n-1}(\mathbf{A}) := \lambda_n(\mathbf{A}), \text{and } \tilde{\lambda}_n(\mathbf{A}) := 0.$$

**Definition 3** (H Quantity). *The following quantities are defined*

$$\tilde{H}_k^{(i)} := \sqrt{k} + \sqrt{N-k}\sqrt{\tilde{\lambda}_{k-1}\left(\boldsymbol{\Sigma}^{(i)}\right)\frac{N}{p\lambda_{\min}^+(\boldsymbol{\Sigma}^{(i)})}},\ \forall k \in [N],$$

*and*

$$\tilde{H}_{\min}^{(i)} := \min_{k \in [N]} \tilde{H}_k^{(i)}, \qquad \tilde{H}_{\min} = \max_{i \in [K]} \tilde{H}_{\min}^{(i)},$$

*where it is clear that* $\tilde{H}_{\min}^{(i)} \leq \tilde{H}_N^{(i)} = \sqrt{N}$.

**Definition 4.** *It is denoted that*

$$G_{\mathcal{S}_1^{(i)},\hat{\boldsymbol{\theta}}^{(i)}} := \tilde{H}_{\min}^{(i)} M_{\mathcal{S}_1^{(i)}}^{(i)} d(\hat{\boldsymbol{\theta}}^{(i)}, \boldsymbol{\theta}^{(i)}), \text{and } G_{\mathcal{S}_1,\hat{\boldsymbol{\theta}}} := \max_{i \in [K]} G_{\mathcal{S}_1^{(i)},\hat{\boldsymbol{\theta}}^{(i)}}.$$

*Here, the subscripts indicate the dependence of* $G$ *on the initial estimator and support estimation of all arms, represented by* $\mathcal{S}_1$ *and* $\hat{\boldsymbol{\theta}}$. $G_{\mathcal{S}_1,\hat{\boldsymbol{\theta}}}$ *is a parameter that can be adaptive to different scenarios using different support estimations* $\{\mathcal{S}_1^{(i)}\}_{i=1}^K$ *and initial estimators* $\{\hat{\boldsymbol{\theta}}\}_{i=1}^K$.

### D.3  THE GENERAL REGRET BOUND

In this section, we begin by stating the key proposition that forms the foundation for the proof of Theorem 1.

**Proposition 6.** *Let* $\boldsymbol{\Gamma}_t^{(i)} = \boldsymbol{\Gamma}_t^{(i)}(\hat{\boldsymbol{\theta}}^{(i)})$. *Under Assumptions 1, 2 and 3, Let* $\hat{\mu}_t^{(i)}$ *be the PWE estimator. Then, with probability at least* $1 - O(1/N)$, *we have:*

$$\left| \hat{\mu}_t^{(i)} - \mu_t^{(i)} \right| \leq C\lambda_N M_{\mathcal{S}_1^{(i)}}^{(i)} \tilde{H}_{\min}^{(i)} d(\hat{\boldsymbol{\theta}}^{(i)}, \boldsymbol{\theta}^{(i)})\mathrm{polylog}(N).$$

We leave the proof of Proposition 6 in Appendix E.

**Remark 6** (Relation to Zhao et al. (2023)). *Zhao et al. (2023) contains limited non-asymptotic results; the component we compare against (PWE for prediction) is asymptotic. In particular, their Theorem 4 is the asymptotic counterpart of Proposition 6:*

$$\left| \hat{\mu}_t^{(i)} - \mu_t^{(i)} \right| = O_p\big(\lambda_N M_{\mathcal{S}_1^{(i)}}^{(i)} \tilde{H}_{\min}^{(i)} d(\hat{\boldsymbol{\theta}}^{(i)}, \boldsymbol{\theta}^{(i)})\big).$$

*The notation $O_p(\cdot)$ in Theorem 4 indicates convergence in probability (asymptotic behavior), whereas Proposition 6 provides a finite-sample, high-probability bound (deterministic $O(\cdot)$ up to a failure probability $O(1/N)$).*

*Proof of Theorem 1.* Our goal is to bound the cumulative regret $R(T)$ of the HOPE algorithm over the time horizon $T$. We decompose the total regret into two parts:

$$R(T) = R_{\text{exploration}} + R_{\text{exploitation}}, \tag{4}$$

where $R_{\text{exploration}}$ is the regret incurred during the exploration phase of length $T_0 = NK$, and $R_{\text{exploitation}}$ is the regret accumulated during the exploitation phase from $T_0 + 1$ to $T$.

During the exploration phase, each arm is pulled exactly $N$ times in a round-robin fashion. At each time step $t$, the regret incurred is at most $\Delta_{\max} := \max_i \sup_t (\mu_t^{(i^*(t))} - \mu_t^{(i)})$, where $\mu_t^{(i^*(t))}$ is the expected reward of the optimal arm at time $t$. Under the boundedness assumption of the reward functions (i.e., $\|\boldsymbol{\theta}^{(i)}\|_2$ and $\|\boldsymbol{x}_t^{(i)}\|$ are bounded), $\Delta_{\max}$ is finite. Therefore, the total regret during the exploration phase is bounded by

$$\mathbb{E}[R_{\text{exploration}}] \leq \sum_{t=1}^{T_0} 2\mathbb{E}\left[ \max_{i \in [K]} \langle \boldsymbol{x}_t^{(i)}, \boldsymbol{\theta}^{(i)} \rangle \right]$$

$$\leq \sum_{t=1}^{T_0} 2\sqrt{c_2} \ (\text{ by Assumption 1 })$$

$$\leq T_0 \times 2\sqrt{c_2}$$

In the exploitation phase, from time $t = T_0 + 1$ to $T$, the algorithm selects the arm with the highest estimated expected reward based on the PWE estimator computed from the exploration data.

The instantaneous regret at time $t$ is $\mu_t^{(i^*(t))} - \mu_t^{(i(t))}$, where $i(t) = \arg\max_i \hat{\mu}_t^{(i)}$ is the arm chosen at time $t$ based on the PWE estimator.

By Proposition 6, the estimation error of the predicted rewards satisfies

$$\left| \hat{\mu}_t^{(i)} - \mu_t^{(i)} \right| \leq C\lambda_N M_{\mathcal{S}_1^{(i)}}^{(i)} \tilde{H}_{\min}^{(i)} d(\hat{\boldsymbol{\theta}}^{(i)}, \boldsymbol{\theta}^{(i)}) \text{polylog}(N),$$

with probability at least $1 - O(1/N)$. Define

$$\epsilon_N := C\lambda_N M_{\mathcal{S}_1^{(i)}}^{(i)} \tilde{H}_{\min}^{(i)} d(\hat{\boldsymbol{\theta}}^{(i)}, \boldsymbol{\theta}^{(i)}) \text{polylog}(N).$$

Let $\mathcal{E}_t$ denote the event that the bound holds for all arms $i \in [K]$ at time $t$:

$$\mathcal{E}_t = \left\{ \forall i \in [K], \forall t \geq T_0 + 1 : \left| \hat{\mu}_t^{(i)} - \mu_t^{(i)} \right| \leq \epsilon_N \right\}.$$

By a union bound over $K$ arms, we have $\Pr(\mathcal{E}_t) \geq 1 - KO(1/N)$.

Under event $\mathcal{E}_t$, the instantaneous regret at time $t \geq T_0 + 1$ is at most $2\epsilon_N$, since $\mu_t^{(i^*(t))} - \epsilon_N \leq \hat{\mu}_t^{(i^*(t))} \leq \hat{\mu}_t^{(i(t))} \leq \mu_t^{(i(t))} + \epsilon_N$, and thus, $\mu_t^{(i^*(t))} - \mu_t^{(i(t))} \leq 2\epsilon_N$.

When $\mathcal{E}_t$ does not occur, the worst-case instantaneous regret is bounded by $\sqrt{c_2}$. Therefore, the expected regret at time $t$ is

$$\mathbb{E}[r_t] \leq 2\epsilon_N \times \Pr(\mathcal{E}_t) + \sqrt{c_2} \times \Pr(\mathcal{E}_t^c).$$

Thus, the expected cumulative regret during the exploitation phase is

$$\mathbb{E}[R_{\text{exploitation}}] \le \sum_{t=T_0+1}^{T} \mathbb{E}[r_t] \le 2(T-T_0)\left(C\lambda_N M_{\mathcal{S}_1^{(i)}}^{(i)}\tilde{H}_{\min}^{(i)}d(\hat{\boldsymbol{\theta}}^{(i)},\boldsymbol{\theta}^{(i)})\text{polylog}(N)+\sqrt{c_2}O(1/N)\right).$$

Combining the exploration and exploitation phases, the total expected regret is

$$\mathbb{E}[R(T)] = \mathbb{E}[R_{\text{exploration}}] + \mathbb{E}[R_{\text{exploitation}}]$$

$$\le 4NK\sqrt{c_2} + 2(T-T_0)\left(C\lambda_N M_{\mathcal{S}_1^{(i)}}^{(i)}\tilde{H}_{\min}^{(i)}d(\hat{\boldsymbol{\theta}}^{(i)},\boldsymbol{\theta}^{(i)})\text{polylog}(N)+\sqrt{c_2}O(1/N)\right)$$

$$= O\left(T_0 + G_{\mathcal{S}_1,\hat{\boldsymbol{\theta}}}(T-T_0)\text{polylog}(T)/\sqrt{N}\right).$$

Thus, we complete the proof. □

### D.4   SCENARIO 1: SPARSE MODEL PARAMETERS

*Proof of Proposition 1.* We establish the regret bound for the HOPE algorithm in the sparse parameter scenario , where the initial estimator $\hat{\boldsymbol{\theta}}_{\text{Lasso}}^{(i)}$ and the support estimator $\mathcal{S}_{1,\text{Lasso}}$ are obtained using Lasso. Under Assumptions 4 and 5 in Appendix G.1, and by applying Proposition 12 and Proposition 13, we obtain the following guarantee:

$$d(\hat{\boldsymbol{\theta}}_{\text{Lasso}}^{(i)},\boldsymbol{\theta}^{(i)}) = O\left(\sqrt{\frac{s_0\log p}{N}}\right), \mathcal{S}_0 \subseteq \mathcal{S}_1, \text{ and } |\mathcal{S}_1| \le C_1|\mathcal{S}_0|, \tag{5}$$

which holds with probability at least $1 - O(1/N)$. This implies that Assumptions 2 and 3 also hold with the same probability. Following the proof technique of Theorem 1, we derive the regret bound for the HOPE algorithm:

$$R(T) = O\left(T_0 + G_{\mathcal{S}_{1,\text{Lasso}},\hat{\boldsymbol{\theta}}_{\text{Lasso}}}(T-T_0)\text{polylog}(N)/\sqrt{N}\right),$$

where $T_0 = NK$ is the length of the exploration phase, and $G_{\mathcal{S}_{1,\text{Lasso}},\hat{\boldsymbol{\theta}}_{\text{Lasso}}}$ is a parameter determined by the Lasso-based support estimation and initial estimator.

The constant $G_{\mathcal{S}_1,\hat{\boldsymbol{\theta}}}$ encapsulates terms arising from the estimation error $d(\hat{\boldsymbol{\theta}}^{(i)},\boldsymbol{\theta}^{(i)})$.

$$G_{\mathcal{S}_1,\hat{\boldsymbol{\theta}}} \le C\max_{i\in[K]}\left(M_{\mathcal{S}_1^{(i)}}^{(i)}\tilde{H}_{\min}^{(i)}d(\hat{\boldsymbol{\theta}}^{(i)},\boldsymbol{\theta}^{(i)})\right)\text{polylog}(T), \tag{6}$$

where $M_{\mathcal{S}_1^{(i)}}^{(i)} = O(\sqrt{s_0})$ and $M_{\mathcal{S}_1^{(i)}}^{(i)}\tilde{H}_{\min}^{(i)} = O(\sqrt{N})$.

Substituting the Lasso estimation error from Equation (5) into $G_{\mathcal{S}_1,\hat{\boldsymbol{\theta}}}$, we obtain the following expression for the total regret $R(T)$ as a function of $N$:

$$R(N) \le C'\left(NK + \frac{(T-NK)\sqrt{s_0\log N}\,\text{polylog}(T)}{\sqrt{N}}\right),$$

where $C'$ is a constant.

With $N$ chosen such that:

$$N^{3/2} = C'T\sqrt{s_0}/K, \tag{7}$$

Substituting $N$ back into the regret expression, the final regret bound becomes:

$$R(T) = O\left(K^{1/3}s_0^{1/3}T^{2/3}\,\text{polylog}(T)\right), \tag{8}$$

which completes the proof. □

### D.5 SCENARIO 2: (APPROXIMATELY) SPARSE EIGENVALUES OF CONTEXT COVARIANCE MATRICES

*Proof of Proposition 2.* We establish the regret bound for the HOPE algorithm in the scenario of approximately sparse eigenvalues , where the initial estimator $\hat{\boldsymbol{\theta}}_{\text{RDL}}^{(i)}$ is used. We don't use the information of sparse model parameters and take $\mathcal{S}_1^{(i)} = [p]$ for each $i \in [K]$.

This analysis applies when the covariance matrix $\boldsymbol{\Sigma}^{(i)}$ for each arm $i \in [K]$ follows the structure outlined in Example 1(A). Under Assumption 6 in Appendix G.1, and by applying Proposition 14, we obtain the following guarantee:

$$d(\hat{\boldsymbol{\theta}}_{\text{RDL}}^{(i)}, \boldsymbol{\theta}^{(i)}) = O\left(\sqrt{T^a/N + T^{-a}}\right), \mathcal{S}_0 \subseteq \mathcal{S}_1, \tag{9}$$

which holds with probability at least $1 - O(1/N)$. This implies that Assumptions 2 and 3 also hold with the same probability. Following the proof technique of Theorem 1, we derive the regret bound for the HOPE algorithm:

$$R(T) = O\left(T_0 + G_{\mathcal{S}_1, \hat{\boldsymbol{\theta}}_{\text{RDL}}}(T - T_0) \, \text{polylog}(N)/\sqrt{N}\right), \tag{10}$$

where $T_0 = NK$ is the length of the exploration phase, and $G_{\mathcal{S}_1, \hat{\boldsymbol{\theta}}_{\text{RDL}}}$ is a parameter determined by the RDL estimator as the initial estimator.

Substituting the RDL estimation error from Equation (9) into $G_{\mathcal{S}_1, \hat{\boldsymbol{\theta}}_{\text{RDL}}}$, we have:

$$G_{\mathcal{S}_1, \hat{\boldsymbol{\theta}}} \leq C \max_{i \in [K]} \left(M_{\mathcal{S}_1^{(i)}}^{(i)} \tilde{H}_{\min}^{(i)} d(\hat{\boldsymbol{\theta}}^{(i)}, \boldsymbol{\theta}^{(i)})\right) \text{polylog}(T). \tag{11}$$

To minimize the total regret $R(T)$, we express the regret as a function of $N$:

$$R(N) \leq C'\left(NK + (T - NK)p^{\frac{1}{T\alpha}}\sqrt{T^a/N + T^{-a}} \, \text{polylog}(T)/\sqrt{N}\right),$$

where $C'$ is a constant.

Let $N$ be chosen such that:

$$N \asymp \left(\max\left\{K^{-\frac{1}{2}}p^{\frac{1}{2T\alpha}}T^{\frac{a+2}{4}}, K^{-\frac{2}{3}}p^{\frac{2}{3T\alpha}}T^{\frac{2-a}{3}}\right\}\right), \tag{12}$$

Substituting $N$ back into the regret expression, the final regret bound becomes:

$$R(T) = \tilde{O}\left(\max\left\{K^{\frac{1}{2}}p^{\frac{1}{2T\alpha}}T^{\frac{a+2}{4}}, K^{\frac{1}{3}}p^{\frac{2}{3T\alpha}}T^{\frac{2-a}{3}}\right\}\right).$$

Thus, we complete the proof. $\qquad\square$

*Proof of Proposition 5.* Similar to proof of Proposition 2 $\qquad\square$

### D.6 SCENARIO 3: BOTH SPARSITIES

**Definition 5.** *We say that the eigenvalues of the covariance matrix decay sufficiently fast if the following condition holds:*

$$\tilde{H}_{\min} \leq O(\sqrt{N}\text{polylog}(T)(s_0 \log p)^{-1/2}) \tag{13}$$

*Proof of Proposition 3.* We aim to establish the regret bound for the HOPE algorithm in the both sparse scenario with the initial estimator $\hat{\boldsymbol{\theta}}_{\text{Lasso}}^{(i)}$.

In the setting where both sparsities are present, each true parameter vector $\boldsymbol{\theta}^{(i)}$ has at most $s_0$ non-zero entries, where $s_0 \ll p$. Additionally, $M_{S_1}\tilde{H}_{\min}$ is either slowly increasing with $N$ or remains bounded.

Recall from Theorem 1 that the regret of the HOPE algorithm is bounded by:

$$R(T) = O\left(T_0 + G_{\mathcal{S}_1, \hat{\boldsymbol{\theta}}}(T - T_0)\, \text{polylog}(N)/\sqrt{N}\right). \tag{14}$$

The constant $G_{\mathcal{S}_1, \hat{\boldsymbol{\theta}}}$ encapsulates terms arising from the estimation error $d(\hat{\boldsymbol{\theta}}_{\text{Lasso}}^{(i)}, \boldsymbol{\theta}^{(i)})$.

$$G_{\mathcal{S}_1, \hat{\boldsymbol{\theta}}} \leq C \max_{i \in [K]} \left(M_{\mathcal{S}_1^{(i)}}^{(i)} \tilde{H}_{\min}^{(i)} d(\hat{\boldsymbol{\theta}}_{\text{Lasso}}^{(i)}, \boldsymbol{\theta}^{(i)})\right) \text{polylog}(T). \tag{15}$$

By substituting Equations (5), (13) and (15) into Equation (14), we arrive at the following expression for the total regret $R(T)$ as a function of $N$:

$$R(N) \leq C'\left(NK + \frac{(T - NK)M\,\text{polylog}(T)}{\sqrt{N}}\right),$$

where $C'$ is a constant and $M$ is defined in Section 6.3.

Choosing $N$ such that:

$$N^{3/2} = \tilde{O}\left(MT/K\right),$$

and substituting $N$ back into the regret expression, the final regret bound becomes:

$$R(T) = \tilde{O}\left(K^{\frac{1}{3}} M^{\frac{2}{3}} T^{\frac{2}{3}}\right),$$

which completes the proof. $\qquad\square$

### D.7 SCENARIO 4: MIXED SPARSITIES

*Proof of Proposition 4.* Refer to the proof of Prop 1 and 2, let $N$ be chosen such that:

$$N \asymp \left(\max\left\{K^{-2/3} s_0^{1/3} T^{2/3}, K^{-\frac{1}{2}} p^{\frac{1}{2T\alpha}} T^{\frac{a+2}{4}}, K^{-\frac{2}{3}} p^{\frac{2}{3T\alpha}} T^{\frac{2-a}{3}}\right\}\right), \tag{16}$$

$$R_T \leq \lambda_N \max_{i \in [K]} \left(M_{\mathcal{S}_1^{(i)}}^{(i)} \tilde{H}_{\min}^{(i)} d(\hat{\boldsymbol{\theta}}^{(i)}, \boldsymbol{\theta}^{(i)})\right) \text{polylog}(T).$$

We now split the maximum over the two parts:

$$R_T \leq \lambda_N \max\left[\max_{i \in \text{Part I}} \left(M_{\mathcal{S}_1^{(i)}}^{(i)} \tilde{H}_{\min}^{(i)} d(\hat{\boldsymbol{\theta}}^{(i)}, \boldsymbol{\theta}^{(i)})\right), \max_{i \in \text{Part II}} \left(M_{\mathcal{S}_1^{(i)}}^{(i)} \tilde{H}_{\min}^{(i)} d(\hat{\boldsymbol{\theta}}^{(i)}, \boldsymbol{\theta}^{(i)})\right)\right] \text{polylog}(T).$$

Referring to the regret bounds for different scenarios in Proposition 1 and Proposition 2, the conclusion follows immediately. $\qquad\square$

## E PROOF OF PROPOSITION 6

In this section, we provide the proof of Proposition 6. To begin, we first establish two lemmas, Lemma 1 and Lemma 2, whose proofs are presented in subsections E.1 and E.2, respectively. These lemmas provide essential intermediary results.

**Lemma 1.** *Under Assumptions 1, 2 and 3, with probability at least $1 - O(1/N)$, it holds that:*

$$\left|\hat{\alpha}_t^{(i)} - \alpha_t^{(i)}\right| \leq C\lambda_N \tilde{H}_{\min}^{(i)} d(\hat{\boldsymbol{\theta}}^{(i)}, \boldsymbol{\theta}^{(i)})\text{polylog}(N).$$

*This result also holds for $S_1 = [p]$.*

**Lemma 2.** *Under Assumptions 1 and 2, there exists a universal constant $C > 0$ such that the following holds. With probability at least $1 - O(1/N)$, for any time index $t$,*

$$\frac{\|\boldsymbol{x}_t^{(i)}\|_2^2}{\|\mathbf{X}^{(i)}\boldsymbol{x}_t^{(i)}\|_2 N^{-1/2}} \leq C \left[ \frac{\mathrm{tr}^2\left(\boldsymbol{\Sigma}_{\mathcal{S}_1^{(i)}}^{(i)}\right)}{\mathrm{tr}\left(\boldsymbol{\Sigma}_{\mathcal{S}_1^{(i)}}^{(i)\,2}\right)} \right]^{1/2} \mathrm{polylog}(N).$$

*Proof of Proposition 6.* The proof follows that of Theorem 4 in Zhao et al. (2023), i.e., an error bound in the form of $O_p$, together with standard concentration inequalities to obtain the explicit high-probability guarantee. By Lemma 1 and 2, we establish the result directly using the following bound:

$$\left|\hat{\mu}_t^{(i)} - \mu_t^{(i)}\right| \leq |\hat{\alpha}_t^{(i)} - \alpha_t^{(i)}| \|\boldsymbol{x}_t^{(i)}\|_2^2 \|\mathbf{X}^{(i)}\boldsymbol{x}_t^{(i)}\|_2^{-1} N^{1/2}$$

$$\leq C\lambda_N \|\boldsymbol{x}_t^{(i)}\|_2^2 \|\mathbf{X}^{(i)}\boldsymbol{x}_t^{(i)}\|_2^{-1} N^{1/2} \tilde{H}_{\min}^{(i)} d(\hat{\boldsymbol{\theta}}^{(i)}, \boldsymbol{\theta}^{(i)}) \mathrm{polylog}(N).$$

$\square$

### E.1 PROOF OF LEMMA 1

To prove Lemma 1, we provide the following lemmas, which provide essential intermediary results.

**Lemma 3.** *Denote $\boldsymbol{\Gamma}_0 = \boldsymbol{\Gamma}_t^{(i)}(\boldsymbol{\theta}^{(i)})$ and $\hat{\boldsymbol{\Gamma}} = \boldsymbol{\Gamma}_t^{(i)}(\hat{\boldsymbol{\theta}}^{(i)}))$ to simplify the notations. Under the assumptions of Lemma 1, it holds*

$$\left|\hat{\alpha}_t^{(i)} - \alpha_t^{(i)}\right| \leq C\lambda_N h(\hat{\boldsymbol{\theta}}^{(i)}) \mathrm{polylog}(N),$$

*with probability at least $1 - N^{-1}$, where $h(\hat{\boldsymbol{\theta}}^{(i)}) = \max\{\|\hat{\boldsymbol{\Gamma}}^{-1}\boldsymbol{\Gamma}_0\|_1, \|\boldsymbol{\Gamma}_0^{-1}\hat{\boldsymbol{\Gamma}}\|_1\}$*

*Proof of Lemma 3.* Same as proof of lemma C.3 in Zhao et al. (2023). $\square$

**Lemma 4.** *Under the assumptions of Lemma 1, it holds that*

$$h(\hat{\boldsymbol{\theta}}^{(i)}) = \max\{\|\hat{\boldsymbol{\Gamma}}^{-1}\boldsymbol{\Gamma}_0\|_1, \|\boldsymbol{\Gamma}_0^{-1}\hat{\boldsymbol{\Gamma}}\|_1\} \leq C\tilde{H}_{\min}^{(i)} d(\hat{\boldsymbol{\theta}}^{(i)}, \boldsymbol{\theta}^{(i)}) \mathrm{polylog}(N),$$

*with probability at least $1 - O(1/N)$.*

*Proof of Lemma 4.* The proof framework is the same as the proof of Lemma 3 in Zhao et al. (2023), except that the corresponding parts are replaced by Lemma 5. $\square$

*Proof of Lemma 1.* The proof of this proposition follows that of Theorem 3 in Zhao et al. (2023), i.e., an error bound in the form of $O_p$, together with standard concentration inequalities to obtain the explicit high-probability guarantee. We can get the proof obviously based on Lemma 3 and Lemma 4.

$\square$

**Lemma 5.** *Under Assumptions 1 and 2, with probability at least $1 - 1/N$, we have*

$$0 < c_l \leq \|N^{-1/2}\mathbf{X}^{(i)}\boldsymbol{\theta}_{\mathbf{Q}_x}^{(i)}\|_2 \leq c_u,$$

$$0 < c_l \leq \|N^{-1/2}\mathbf{X}^{(i)}\hat{\boldsymbol{\theta}}_{\mathbf{Q}_x}^{(i)}\|_2 \leq c_u,$$

*where $c_l$ and $c_u$ are constants.*

*Proof of Lemma 5.* For the sake of notational simplicity, we omit the superscript $(i)$ in this proof.

*Step 1. Conditioning on $\boldsymbol{x}$*

$$\|N^{-1/2}\mathbf{X}\boldsymbol{\theta}_{\mathbf{Q}_x}\| = \|N^{-1/2}\mathbf{X}(\mathbf{I} - \frac{\boldsymbol{x}^\top \boldsymbol{x}}{\|\boldsymbol{x}\|_2^2})\boldsymbol{\theta}\|$$

$\boldsymbol{\theta}_{\mathbf{Q}_x} = \boldsymbol{\theta} - \frac{\boldsymbol{x}^\top \boldsymbol{\theta}}{\|\boldsymbol{x}\|_2^2} \boldsymbol{x}$ Since the entries $\boldsymbol{x}_\tau^\top \boldsymbol{\theta}_{\mathbf{Q}_x}$ are i.i.d. from $N(0, \boldsymbol{\theta}_{\mathbf{Q}_x}^\top \boldsymbol{\Sigma} \boldsymbol{\theta}_{\mathbf{Q}_x})$, we have

$$\|\mathbf{X}\boldsymbol{\theta}_{\mathbf{Q}_x}\|_2^2 = \sum_{\tau=1}^{N} (\boldsymbol{x}_\tau^\top \boldsymbol{\theta}_{\mathbf{Q}_x})^2 =^{\mathrm{d}} (\boldsymbol{\theta}_{\mathbf{Q}_x}^\top \boldsymbol{\Sigma} \boldsymbol{\theta}_{\mathbf{Q}_x}) \chi_N^2$$

A $\chi_N^2$ random variable with $N$ degrees of freedom concentrates around $N$ via standard tail bounds:

$$\Pr\left[ |\|\chi_N^2 \geq \delta N \right] \leq 2\exp(-cN\min(\delta, \delta^2)),$$

for some absolute constant $c > 0$. Setting $\delta = C_0 \frac{\log N}{N} \leq 1 - \alpha$ where $\alpha$ is a constant. One can get:

$$\Pr\left[ (1-\delta)N \leq \chi_N^2 \leq (1+\delta)N \right] \geq 1 - \frac{C_1}{N},$$

Hence, conditioned on $\boldsymbol{\theta}_{\mathbf{Q}_x}$, with probability at least $1 - \frac{C_1}{N}$,

$$(1-\delta)N(\boldsymbol{\theta}_{\mathbf{Q}_x}^\top \boldsymbol{\Sigma} \boldsymbol{\theta}_{\mathbf{Q}_x}) \leq \|\mathbf{X}\boldsymbol{\theta}_{\mathbf{Q}_x}\|_2^2 \leq (1+\delta)N(\boldsymbol{\theta}_{\mathbf{Q}_x}^\top \boldsymbol{\Sigma} \boldsymbol{\theta}_{\mathbf{Q}_x}).$$

Taking square-roots and dividing by $\sqrt{N}$,

$$\sqrt{(1-\delta)}\sqrt{\boldsymbol{\theta}_{\mathbf{Q}_x}^\top \boldsymbol{\Sigma} \boldsymbol{\theta}_{\mathbf{Q}_x}} \leq \|N^{-1/2}\mathbf{X}\boldsymbol{\theta}_{\mathbf{Q}_x}\|_2 \leq \sqrt{(1+\delta)}\sqrt{\boldsymbol{\theta}_{\mathbf{Q}_x}^\top \boldsymbol{\Sigma} \boldsymbol{\theta}_{\mathbf{Q}_x}}.$$

*Step 2. Concentration of $\boldsymbol{\theta}_{\mathbf{Q}_x}^\top \boldsymbol{\Sigma} \boldsymbol{\theta}_{\mathbf{Q}_x}$*

$$\boldsymbol{\theta}_{\mathbf{Q}_x}^\top \boldsymbol{\Sigma} \boldsymbol{\theta}_{\mathbf{Q}_x} = \boldsymbol{\theta}^\top \boldsymbol{\Sigma} \boldsymbol{\theta} + \frac{(\boldsymbol{x}^\top \boldsymbol{\theta})^2}{\|\boldsymbol{x}\|_2^4} \boldsymbol{x}^\top \boldsymbol{\Sigma} \boldsymbol{x} - 2\frac{\boldsymbol{x}^\top \boldsymbol{\theta}}{\|\boldsymbol{x}\|^2} \boldsymbol{\theta}^\top \boldsymbol{\Sigma} \boldsymbol{x}$$

$$\boldsymbol{\theta}_{\mathbf{Q}_x} = \boldsymbol{\theta} - <\boldsymbol{\theta}, \frac{\boldsymbol{x}}{\|\boldsymbol{x}\|_2}> \frac{\boldsymbol{x}}{\|\boldsymbol{x}\|_2} \tag{17}$$

In high dimensional case, when $\boldsymbol{x} \sim \boldsymbol{\Sigma}$ and $\boldsymbol{\Sigma}$ satisfies some mild conditions.

By assumptions, the following three intermediate results are what we need

$$\Pr(\boldsymbol{\theta}^\top \boldsymbol{x} \leq \log N * c_2) \geq 1 - \frac{1}{N} \tag{18}$$

$$\Pr(\|\boldsymbol{x}\|_2 \leq \sqrt{p}/2) \geq 1 - \frac{1}{N} \tag{19}$$

$$\Pr(\|\boldsymbol{\theta}\|_2 \geq \frac{\log N}{\sqrt{p}}) = 1. \tag{20}$$

So we have a constant $\alpha \in (0, 1/3)$, so that with probability at least $1 - 1/N$:

$$\Pr\left[ |\boldsymbol{x}^\top \boldsymbol{\theta}| > \alpha \|\boldsymbol{\theta}\| \|\boldsymbol{x}\| \right] \leq 1/N \quad \text{for some } c_0 > 0.$$

So with probability at least $1 - 1/N$,

$$\boldsymbol{\theta}_{\mathbf{Q}_x}^\top \boldsymbol{\Sigma} \boldsymbol{\theta}_{\mathbf{Q}_x} \geq \alpha_0 \boldsymbol{\theta}^\top \boldsymbol{\Sigma} \boldsymbol{\theta},$$

for some constant $\alpha_0 > 0$ depending on $\alpha$ and on the spectral properties of $\boldsymbol{\Sigma}$.

*Step 3. Concentration of $\|N^{-1/2}\mathbf{X}\boldsymbol{\theta}_{\mathbf{Q}_x}\|_2$* By a union bound, with probability at least $1 - 1/N$, we have

$$\sqrt{(1-\delta)}\sqrt{\alpha_0}\sqrt{\boldsymbol{\theta}^\top \boldsymbol{\Sigma} \boldsymbol{\theta}} \leq \|N^{-1/2}\mathbf{X}\boldsymbol{\theta}_{\mathbf{Q}_x}\|_2 \leq \sqrt{(1+\delta)}\sqrt{1+\alpha_0}\sqrt{\boldsymbol{\theta}^\top \boldsymbol{\Sigma} \boldsymbol{\theta}}.$$

Then we can get with probability at least $1 - 1/N$, we have

$$0 < c_l \leq \|N^{-1/2}\mathbf{X}\boldsymbol{\theta}_{\mathbf{Q}_x}\|_2 \leq c_u,$$

where $c_l$ and $c_u$ are constants.

*Step 4. Concentration of $\|N^{-1/2}\mathbf{X}\hat{\boldsymbol{\theta}}_{\mathbf{Q}_x}\|_2$*

By Assumption 2, the estimation error satisfies

$$0 \leq \|\hat{\boldsymbol{\theta}}^\top \boldsymbol{\Sigma} \hat{\boldsymbol{\theta}} - \boldsymbol{\theta}^\top \boldsymbol{\Sigma} \boldsymbol{\theta}\| \leq d \leq \min(c_1, c_2)/2 \tag{21}$$

with probability at least $1 - 1/N$.

Thus, with probability at least $1 - 1/N$:

$$\|\hat{\boldsymbol{\theta}}^\top \boldsymbol{\Sigma} \hat{\boldsymbol{\theta}}\| \geq c_1 - d \geq c_1/2, \tag{22}$$

Then we have that with probability at least $1 - 1/N$,

$$\|\hat{\boldsymbol{\theta}}\|_2^2 \geq \frac{1}{\lambda_{\max}} |\hat{\boldsymbol{\theta}}^\top \boldsymbol{\Sigma} \hat{\boldsymbol{\theta}}| \overset{(22)}{>} \frac{1}{\lambda_{\max}} \frac{c_1}{2} \overset{\lambda_{\max} \leq \frac{p}{\log N}}{\geq} \frac{\log N}{p}.$$

Following steps analogous to Steps 1-3, we conclude that with probability at least $1 - 1/N$,

$$0 < c_l \leq \|N^{-1/2} \mathbf{X} \hat{\boldsymbol{\theta}}_{\mathbf{Q}_x}\|_2 \leq c_u,$$

This completes the proof.

$\square$

### E.2 PROOF OF LEMMA 2

*Proof of Lemma 2.* For the sake of notational simplicity, we omit the subscript: $\mathcal{S}_1^{(i)}$.

*Step 1. Boundedness of $\|\boldsymbol{x}_t^{(i)}\|_2^2$.*

We have:

$$\mathbb{E}\big[\|\boldsymbol{x}_t^{(i)}\|_2^2\big] \;=\; tr(\boldsymbol{\Sigma}^{(i)}).$$

Using the fact that a Gaussian vector $\boldsymbol{x}_t^{(i)} \sim \mathcal{N}(0, \boldsymbol{\Sigma}^{(i)})$ admits the representation $\boldsymbol{x} = \Sigma^{1/2} \boldsymbol{z}$ with $\boldsymbol{z} \sim \mathcal{N}(0, I)$, one finds

$$\|\boldsymbol{x}_t^{(i)}\|_2^2 \;=\; \|\boldsymbol{\Sigma}^{(i)1/2} \mathbf{z}\|_2^2 \;=\; \sum_{j=1}^p \lambda_j \, (z_j^2),$$

where $\lambda_1, \ldots, \lambda_p$ are the eigenvalues of $\boldsymbol{\Sigma}^{(i)}$. Since $\mathbb{E}[z_j^2] = 1$, we obtain $\mathbb{E}[\|\boldsymbol{x}_t^{(i)}\|_2^2] = tr(\boldsymbol{\Sigma}^{(i)})$. Furthermore, $\chi^2$ concentration inequality ensures that for *high probability* (at least $1 - O(1/N)$),

$$\big|\|\boldsymbol{x}_t^{(i)}\|_2^2 - tr(\boldsymbol{\Sigma}^{(i)})\big| \;\leq\; \delta \, tr(\boldsymbol{\Sigma}^{(i)}),$$

where $\delta > 0$ can be chosen so that $\exp(-cp\,\delta^2) \approx 1/N$, thus $\delta$ scales roughly like $\sqrt{(\log N)/p}$. Hence we can absorb the deviation factor into a $\mathrm{polylog}(N)$ term (and a universal constant). Concretely,

$$\|\boldsymbol{x}_t^{(i)}\|_2^2 \;\leq\; (1 + \delta) \, tr(\boldsymbol{\Sigma}^{(i)}) \;\leq\; C_1 \, tr\big(\boldsymbol{\Sigma}^{(i)}\big) \, \mathrm{polylog}(N),$$

for some absolute constant $C_1 > 0$ and with probability at least $1 - O\big(\frac{1}{N}\big)$.

*Step 2. Controlling $\|\mathbf{X}^{(i)} \boldsymbol{x}_t^{(i)}\|_2 / \sqrt{N}$.*

Condition on the vector $\boldsymbol{x}_t^{(i)}$. By our assumptions, *each row* of $\mathbf{X}^{(i)}$ is drawn from $N(\mathbf{0}, \boldsymbol{\Sigma}^{(i)})$, independently of other rows. Let $\boldsymbol{x}_\tau^\top$ be the $i$-th row of $\mathbf{X}^{(i)}$. Then, for a *fixed* $\boldsymbol{x}_t^{(i)}$, we observe:

$$\boldsymbol{x}_\tau^\top \boldsymbol{x}_t^{(i)} \;\sim\; N\Big(0, \; \boldsymbol{x}_t^{(i)\top} \boldsymbol{\Sigma}^{(i)} \boldsymbol{x}_t^{(i)}\Big).$$

In other words, each scalar $\boldsymbol{x}_\tau^\top \boldsymbol{x}_t^{(i)}$ is a Gaussian with variance $\boldsymbol{x}_t^{(i)\top} \boldsymbol{\Sigma}^{(i)} \boldsymbol{x}_t^{(i)}$, and these $N$ scalars are i.i.d. given $\boldsymbol{x}_t^{(i)}$. Hence,

$$\|\mathbf{X}^{(i)} \boldsymbol{x}_t^{(i)}\|_2^2 \;=\; \sum_{i=1}^N \big(\mathbf{z}_i^\top \boldsymbol{x}_t^{(i)}\big)^2 \;\overset{d}{=}\; \big(\boldsymbol{x}_t^{(i)\top} \boldsymbol{\Sigma}^{(i)} \boldsymbol{x}_t^{(i)}\big) \, \chi_N^2,$$

where $\chi_N^2$ denotes a chi-square random variable with $N$ degrees of freedom.

*Concentration argument.* A $\chi_N^2$ variable concentrates around $N$, so with high probability,

$$\chi_N^2 \; \approx \; N \; \left(1 \pm O(\tfrac{1}{\sqrt{N}})\right).$$

Therefore, with probability at least $1 - O\left(\tfrac{1}{N}\right)$,

$$\|\mathbf{X}^{(i)} \, \boldsymbol{x}_t^{(i)}\|_2^2 \; = \; \left(\boldsymbol{x}_t^{(i)\top} \, \boldsymbol{\Sigma}^{(i)} \, \boldsymbol{x}_t^{(i)}\right) \chi_N^2 \; \leq \; C \, N \, \boldsymbol{x}_t^{(i)\top} \, \boldsymbol{\Sigma}^{(i)} \, \boldsymbol{x}_t^{(i)},$$

for some absolute constant $C > 0$. Taking square roots gives the desired statement on $\|\mathbf{X}^{(i)} \, \boldsymbol{x}_t^{(i)}\|_2 / \sqrt{N}$.

By Hanson-Wright inequality:

$$\Pr(\boldsymbol{x}_t^{(i)\top} \boldsymbol{\Sigma}^{(i)} \boldsymbol{x}_t^{(i)} - \mathbb{E}(\boldsymbol{x}_t^{(i)\top} \boldsymbol{\Sigma}^{(i)} \boldsymbol{x}_t^{(i)}) < -t) \leq \exp\left(-c \min\left(\frac{t^2}{\|\boldsymbol{\Sigma}^{(i)}\|_F^2}, \frac{t}{\|\boldsymbol{\Sigma}^{(i)}\|_2}\right)\right).$$

We let $\|\boldsymbol{\Sigma}^{(i)}\|_F \sqrt{\frac{\ln N}{c}} \leq t \leq \frac{\|\boldsymbol{\Sigma}^{(i)}\|_F^2}{\|\boldsymbol{\Sigma}^{(i)}\|_2}$, which is well defined by Assumption 1.Then, we get

$$\Pr\left(\boldsymbol{x}^\top \boldsymbol{\Sigma}^{(i)} \boldsymbol{x} - tr(\boldsymbol{\Sigma^2})\right) < \|\boldsymbol{\Sigma}^{(i)}\|_F \sqrt{\frac{\ln N}{c}}) \leq 1/N.$$

$$\Pr\left(\boldsymbol{x}^\top \boldsymbol{\Sigma}^{(i)} \boldsymbol{x} < tr(\boldsymbol{\Sigma^2}) \left(1 - \frac{1}{\|\boldsymbol{\Sigma}^{(i)}\|_F} \sqrt{\frac{\ln N}{c}}\right)\right) \leq 1/N.$$

So

$$\Pr\left(\boldsymbol{x}^\top \boldsymbol{\Sigma}^{(i)} \boldsymbol{x} < tr(\boldsymbol{\Sigma}^{(i)^2})\tau\right) \leq 1/N.$$

where $0 < \tau < 1$. Thus, in large-dimension or sub-Gaussian cases, $\boldsymbol{x}_t^{(i)\top} \Sigma^{(i)} \boldsymbol{x}_t^{(i)}$ itself is on the order of $\mathrm{tr}\left((\Sigma^{(i)})^2\right)$, up to the usual polylog factors. Therefore,

$$\|\mathbf{X}^{(i)} \, \boldsymbol{x}_t^{(i)}\|_2 / \sqrt{N} \; \geq \; \frac{1}{C_2} \sqrt{\mathrm{tr}\left(\left(\Sigma^{(i)}\right)^2\right)} / \mathrm{polylog}(N)$$

with probability at least $1 - O\left(\tfrac{1}{N}\right)$, where $C_2 > 0$ is another universal constant.

*Step 3. Conclude the ratio bound.* Combining the above:

$$\frac{\|\boldsymbol{x}_t^{(i)}\|_2^2}{\|\mathbf{X}^{(i)} \, \boldsymbol{x}_t^{(i)}\|_2 / \sqrt{N}} \; \leq \; \frac{C_1 \, \mathrm{tr}\left(\Sigma^{(i)}\right) \mathrm{polylog}(N)}{\frac{1}{C_2} \sqrt{\mathrm{tr}\left(\left(\Sigma^{(i)}\right)^2\right)} / \mathrm{polylog}(N)} \; = \; \left(C_1 \, C_2\right) \sqrt{\frac{\left(\mathrm{tr}(\Sigma^{(i)})\right)^2}{\mathrm{tr}\left(\left(\Sigma^{(i)}\right)^2\right)}} \, [\mathrm{polylog}(N)]^2.$$

We absorb $[\mathrm{polylog}(N)]^2$ into a single $\mathrm{polylog}(N)$ factor, and set $C = C_1 \, C_2$ (both are universal constants). Hence, with probability at least $1 - O(1/N)$,

$$\frac{\|\boldsymbol{x}_t^{(i)}\|_2^2}{\|\mathbf{X}^{(i)} \, \boldsymbol{x}_t^{(i)}\|_2 \, N^{-1/2}} \; \leq \; C \sqrt{\frac{\mathrm{tr}^2\left(\boldsymbol{\Sigma}^{(i)}\right)}{\mathrm{tr}\left(\left(\boldsymbol{\Sigma}^{(i)}\right)^2\right)}} \, \mathrm{polylog}(N).$$

This completes the proof. $\qquad\qquad\qquad\qquad\qquad\qquad\qquad\qquad\qquad\qquad\qquad\qquad\qquad\square$

## F    ADDITIONAL DISCUSSION ON PARAMETER-AWARENESS

In this section, we present the theoretical results for the agnostic version of HOPE across the four scenarios.

**Proposition 7** (Sparse Model Parameters (parameter-agnostic version)). *With Lasso as the initial estimator and also Lasso to perform the support estimation, using $N \asymp K^{-2/3}T^{2/3}$, under Assumption 1 and the conditions in Appendix G.1, G.2 for the guarantee of Lasso, the regret of HOPE is bounded as*

$$R(T) = O\big(K^{\frac{1}{3}} s_0^{\frac{1}{2}} T^{\frac{2}{3}} \mathrm{polylog}(T)\big).$$

*Proof of Proposition 7.* Similar to proof of Proposition 1 in Section D.4. □

**Proposition 8** (Sparse Eigenvalues of $\boldsymbol{\Sigma}$: Example 1(A) (parameter-agnostic version)). *With RDL as the initial estimator and $\mathcal{S}_1^{(i)} = [p]$ for all arms, using $N \asymp \max\{K^{-\frac{1}{2}}T^{\frac{1}{2}}, K^{-\frac{2}{3}}T^{\frac{1}{3}}\}$ under Assumption 1 and the conditions in Appendix G.3 for the guarantee of RDL, if the covariance matrices satisfy Example 1(A), the regret of HOPE is bounded as*

$$R(T) = \tilde{O}\big( \max \big\{ K^{\frac{1}{2}} p^{\frac{1}{T^a}} T^{\frac{a+1}{2}}, K^{\frac{1}{3}} p^{\frac{1}{T^a}} T^{\frac{3-2a}{3}} \big\}\big).$$

*Proof of Proposition 8.* Similar to proof of Proposition 2 in Section D.5. □

**Proposition 9** (Sparse Eigenvalues of $\boldsymbol{\Sigma}$: Example 1(B) (parameter-agnostic version)). *With RDL as the initial estimator and $\mathcal{S}_1^{(i)} = [p]$ for all arms, using $N \asymp K^{-\frac{1}{2}}T^{\frac{1}{2}}$, under Assumption 1 and the additional conditions specified in Appendix G.3 for the guarantee of RDL, if the covariance matrices satisfy Example 1(B), the regret of HOPE is bounded as*

$$R(T) = \tilde{O}\big(K^{\frac{1}{2}} T^{\frac{1}{2} + \frac{3c(1-b)}{2}}\big).$$

*Proof of Proposition 9.* Similar to proof of Proposition 5 in Section D.5. □

**Proposition 10** (Both Sparsities (parameter-agnostic version)). *With Lasso as the initial estimator and also Lasso to perform the support estimation, using $N \asymp K^{-2/3}T^{2/3}$, under Assumption 1 and the conditions in Appendix G.1, G.2 for the guarantees of Lasso, if the eigenvalues of covariance matrices $\boldsymbol{\Sigma}^{(i)}$ for all $i \in [K]$ decay sufficiently fast (e.g., Example 1; see Appendix D.6 for details), the regret of HOPE is bounded as*

$$R(T) = \tilde{O}\big(K^{\frac{1}{3}} M T^{\frac{2}{3}}\big).$$

**Proposition 11** (Mixed Sparsity). *With Lasso as the initial estimator and also Lasso to perform the support estimation for arms in Part I, and RDL as the initial estimator and $\mathcal{S}_1^{(i)} = [p]$ for arms in Part II, using $N \asymp K^{-2/3}T^{2/3}$, under Assumption 1 and the conditions in Appendix G.1, G.2, G.3 for the guarantees of Lasso and RDL, if the covariance matrices of arms in Part II satisfies Example 1(A), the regret of HOPE is bounded as*

$$R(T) = \tilde{O}\big( \max \big\{ K^{\frac{1}{3}} s_0^{\frac{1}{3}} T^{\frac{2}{3}}, K^{\frac{1}{2}} p^{\frac{1}{T^a}} T^{\frac{a+1}{2}}, K^{\frac{1}{3}} p^{\frac{1}{T^a}} T^{\frac{3-2a}{3}} \big\}\big).$$

## G    ADDITIONAL TECHNIQUES

### G.1    THEORY FOR LASSO PREDICTION ERROR

In this section, we derive a theoretical upper bound for the prediction error of the Lasso estimator in a high-dimensional linear regression setting.

**Assumption 4.** *1. There exists a constant $\kappa > 0$ such that for all vectors $\boldsymbol{\delta} \in \mathbb{R}^p$ satisfying $\|\boldsymbol{\delta}_{\mathcal{S}_0^c}\|_1 \leq 3\|\boldsymbol{\delta}_{\mathcal{S}_0}\|_1$, where $\mathcal{S}_0$ is the support of $\boldsymbol{\theta}$ with $|\mathcal{S}_0^{(i)}| \leq s_0$, the following holds: $\frac{1}{N}\|\mathbf{X}^{(i)}\boldsymbol{\delta}\|_2^2 \geq \kappa\|\boldsymbol{\delta}_S\|_2^2$.*

*2. $\lambda \asymp \sigma\sqrt{\frac{\log p}{N}}$, where $\sigma^2$ is the variance of the noise.*

**Remark 7.** *Assumption 4 is invoked in Propositions 1, 3, and 4 to provide a standard framework for analyzing Lasso estimation performance, as illustrated in Proposition 12. We stress that this assumption is solely employed to derive the Lasso estimator's error bound in Proposition 12, which subsequently propagates to Propositions 1, 3, and 4. Beyond this, Assumption 4 plays no further role in the theoretical analysis of these propositions. The literature presents well-established alternatives for deriving Lasso error bounds, such as the Restricted Eigenvalue (RE) condition or the compatibility condition, widely adopted in Lasso bandit studies (Li et al. (2022); Bastani & Bayati (2020)). Any of these conditions could readily replace Assumption 4 without affecting our core results. Crucially, since the primary contribution of our methodology lies in the PWE algorithm framework—where Lasso serves merely as one possible initialization tool—the specific assumptions governing Lasso's estimation properties are peripheral to our theoretical focus. A parallel argument applies to Assumption 5: its sole purpose is to enable the support recovery guarantees in Proposition 13, and it plays no further role in the proofs or conclusions of the aforementioned propositions. Notably, various support estimation methods (e.g., SISFan & Lv (2008), KnockoffCandes et al. (2018)) exist, each with their own standard conditions to achieve the desired theoretical guarantees. These could readily substitute Assumption 5, but as this lies beyond the scope of our work, we omit further discussion.*

**Proposition 12.** *Under Assumptions 4, we have*

$$d(\hat{\boldsymbol{\theta}}_{Lasso}^{(i)}, \boldsymbol{\theta}^{(i)}) = O\left(\sqrt{\frac{s_0 \log p}{N}}\right), \tag{23}$$

*Proof of Proposition 12.* By Section 2.4 of Bühlmann & Van De Geer (2011). □

### G.2 THEORY OF SUPPORT ESTIMATION WITH LASSO

In this section, we present a theoretical analysis of the Lasso estimator's ability to perform variable selection in high-dimensional linear regression models. We establish conditions under which the Lasso is able to provide a good estimate of the true support set of the parameter vector $\boldsymbol{\theta}^{(i)}$. The analysis is based on classical assumptions and leverages key results from the literature.

**Assumption 5.** $\theta_{\min}^{(i)} = \min_{j \in \mathcal{S}_0^{(i)}} |\theta_j^{(i)}|$ *satisfies* $\theta_{\min}^{(i)} \geq C\sigma\sqrt{\frac{\log p}{N}}$

**Remark 8.** *Unlike our approach, the method proposed by Li et al. (2022) cannot directly incorporate support estimation. Consequently, their framework cannot leverage Assumption 5 to achieve improved performance guarantees.*

**Proposition 13.** *Under Assumptions 4 and 5, there exist a constant $C_1$, with probability at least $1 - O(1/N)$, the support estimation by Lasso satisfies the following:*

$$\mathcal{S}_0^{(i)} \subseteq \mathcal{S}_1^{(i)}, \text{ and } |\mathcal{S}_1^{(i)}| \leq C_1 |\mathcal{S}_0^{(i)}|$$

*Proof of Proposition 13.* In Section 2.4 of Bühlmann & Van De Geer (2011). □

### G.3 THEORY FOR RDL PREDICTION ERROR

Fix a covariance matrix $\boldsymbol{\Sigma}^{(i)}$ with eigenvalues $\left\{\lambda_k^{(i)}\right\}_{k \in [p]}$. We create two sequences called effective bias/variance denoted as $B_{N,T}^{(i)}$ and $V_{N,T}^{(i)}$, based on a budget of $T$ and the number of samples $N$ used for estimation.

$$B_{N,T}^{(i)} := \lambda_{k^*}^{(i)}, \text{ and } V_{N,T}^{(i)} := \left(\frac{k^*}{N} + \frac{N}{R_{k^*}\left(\Sigma^{(i)}\right)}\right)$$

For $k \in [p]$, we define an empirical submatrix as $\mathbf{X}_{k+1:p}^{(i)} \in \mathbb{R}^{N \times (p-k)}$ as the $p - k$ columns to the right of $\mathbf{X}^{(i)}$, and define a Gram sub-matrix $A_k^{(i)} = \mathbf{X}_{k+1:p}^{(i)} \left(\mathbf{X}_{k+1:p}^{(i)}\right)^{\top} \in \mathbb{R}^{N \times N}$.

**Assumption 6.** *There exist $c_U > 1$ such that $k_N^* < N/c_U$ and a conditional number of $A_k^{(i)}$ is positive with probability at least $1 - c_U e^{-N/c_U}$,*

**Remark 9.** *This standard assumption, following Komiyama & Imaizumi (2024), is only used to derive the prediction error bound for the RDL estimator in Proposition 14. While this bound is subsequently utilized in Propositions 2, 5, and 4, we emphasize that the assumption itself is not invoked anywhere else in these proofs or in our theoretical framework.*

**Proposition 14.** *Under Assumptions 6, we have*

$$d(\hat{\boldsymbol{\theta}}_{RDL}^{(i)}, \boldsymbol{\theta}^{(i)}) \leq C_U \left( B_{N,T}^{(i)} + V_{N,T}^{(i)} \right), \tag{24}$$

*with some constant $C_U > 0$ and probability at least $1 - 2c_U e^{-N/c_U}$.*

*Proof of Proposition 14.* By Theorem 2 in Komiyama & Imaizumi (2024) □

## H  BROADER IMPACTS

This work introduces a novel approach, HOPE, to high-dimensional linear contextual bandit problems, which adapts to both sparse model parameters and sparse eigenvalues of context covariance matrices. This advance addresses significant challenges in high-dimensional bandit problems, by offering a more flexible and generalizable method compared to existing techniques. We do not foresee major negative societal impacts, as the work primarily focuses on advancing theoretical methods in the field.

## I  LIMITATION AND FUTURE WORK

While the proposed HOPE algorithm, built upon PWE and ETC frameworks, demonstrates promising results across various scenarios, particularly achieving sublinear regret for the first time in mixed scenarios, there remain several promising avenues for future research.

- *Linearity Assumption*: Our theoretical guarantees and empirical results are restricted to linear high-dimensional settings. However, real-world reward structures often exhibit nonlinear patterns. Extending pointwise estimation to nonlinear models (e.g., kernel methods or neural networks) could expand the applicability of HOPE. Developing corresponding regret analyses in these settings is a critical and challenging next step.

- *Exploration Strategy*: HOPE currently employs an Explore-Then-Commit (ETC) strategy, which, while effective, may not fully exploit the advantages of adaptive exploration. Integrating pointwise estimation with adaptive methods such as UCB or Thompson Sampling could improve learning efficiency. Notably, the confidence intervals derived from pointwise estimation may naturally align with these adaptive strategies to yield tighter regret bounds.

- *Broader Reinforcement Learning Applications*: The effectiveness of pointwise estimation in contextual bandits suggests potential for broader use in reinforcement learning (RL), particularly in contextual MDPs where partial feedback and high-dimensional state representations are common. Extending HOPE to RL domains could bridge insights between bandit theory and sequential decision-making under uncertainty.

## J  USE OF LARGE LANGUAGE MODELS

We used a large language model (e.g., ChatGPT) solely as a writing-assistance tool for minor grammar, style, and wording edits of author-written text. No parts of the research, including theoretical results, were materially influenced by LLMs. All content was authored and verified by the authors, who take full responsibility for the manuscript.

## K  ETHICS STATEMENT

This paper focuses on theoretical analysis and algorithmic development for high-dimensional contextual bandits. All results are derived under formal mathematical assumptions and validated using synthetic simulations only. No sensitive, personal, or proprietary data is used. We therefore emphasize that our contributions are purely methodological, and any downstream application should incorporate domain-specific ethical guidelines.

## L    REPRODUCIBILITY STATEMENT

We have taken several steps to ensure reproducibility:

- **Theoretical results:** All assumptions, lemmas, and proofs are provided in the main text and appendices.
- **Algorithmic description:** Pseudocode for HOPE and its variants is included, with full specification of parameter choices.
- **Simulation setup:** Details of synthetic data generation, parameter settings, and evaluation metrics are described in Section 7.
- **Code availability:** We will release our implementation, including data generation scripts and evaluation pipelines, in a public repository upon publication.

Together, these materials enable independent verification of both our theoretical and empirical results.

