# OpenReview forum: "Navigating Sparsities in High-Dimensional Linear Contextual Bandits"
_ICLR.cc/2026/Conference — Submitted to ICLR 2026_

### Official Review · Reviewer_6few · 2025-10-23

**Soundness:** 3
**Presentation:** 2
**Contribution:** 2
**Rating:** 4
**Confidence:** 3

**Summary:**

This paper studies high-dimensional linear contextual bandits, a challenging problem due to the curse of dimensionality.
Since the general high-dimensional case is unsolvable, existing research has introduced additional structural assumptions, typically the sparsity of either the model parameter or the covariance matrix of the context distribution.
Under the sparse-parameter assumption, the LASSO estimator is commonly used, while a recent work employs the RDL estimator for cases with sparse covariance matrices.
However, previous studies can handle only one of these cases at a time and cannot address hybrid/both settings.
To overcome this limitation, this paper adopts the PWE and proposes a policy called HOPE, an EtC-type policy.

**Strengths:**

* The proposed PWE-based policy can handle both cases: (i) a sparse model parameter and (ii) a sparse covariance matrix of the context distribution. In contrast, the LASSO estimator can be applied only to (i), and the RDL estimator only to (ii). Moreover, the proposed HOPE policy is applicable to both Scenarios 3 and 4.
* The paper provides a potentially tighter regret bound under Scenario 3 by incorporating the effective rank of the covariance matrix.

**Weaknesses:**

* Unlike previous EtC-type policies for high-dimensional linear contextual bandits—such as those by Hao et al. (2020) and Li et al. (2022), which adopt the LASSO estimator for case (i), and Komiyama & Imaizumi (2024), which adopts the RDL estimator for case (ii)—the proposed HOPE policy appears to require computing the LASSO estimator at every exploitation round with dimension $N+1$. Since the theoretical results (Propositions 1–4) involve the choice of $N$, even at the order of $T^{2/3}$, the proposed method incurs additional computational cost, which is expected to be substantial for large $T$ (as it computes LASSO estimator in $T^{2/3}+1$ dimension at least $T-T_0$ times), compared with existing EtC approaches that compute the estimator only once. Although numerical results are provided (with small $T$), the performance improvement appears marginal. Therefore, a more detailed comparison of computational costs would be valuable.
* It is difficult to compare the results with prior works, particularly for Scenario 2, where certain instance-dependent constants $a$ and $c$ are introduced without explanation. While the authors claim that the corresponding proposition yields an improved regret rate, this is not clear from the current presentation without clarification of the role of $c$.

**Questions:**

1. In Scenario 4, it appears that the policy knows in advance which arms belong to Part 1 or Part 2. While the authors claim that none of the previous works can handle Scenario 4, one might also consider a simple EtC-type approach that applies the LASSO estimator to Part 1 (where the dimension of $\theta$ is truncated to the size of Part 1) and the RDL estimator to Part 2. Since both estimators exhibit good concentration properties, such an EtC policy may achieve a similar regret bound under a simple worst-case analysis. Could the authors clarify the potential advantages of the proposed approach over this simple alternative, particularly in the context of Scenario 4?

2. Related to Weakness 1: Could the authors provide a detailed runtime comparison between the proposed HOPE policy and existing EtC-type policies?

3. Related to Weakness 2: While $\alpha,a,b$ and $c$ are instance-dependent constants, are there any known bounded relationships or ranges for these parameters? For instance, is $b\in (0,1)$ or are $a$ and $c$ related in a specific way?

4. Could the authors elaborate on Lines 849–850 (“to mitigate … where $i_0$ ...")? It seems that the authors replace one column with $\bar{\zeta}$ (based on the LASSO estimator) to further exploit sparsity in the model parameter. However, the intended meaning of these lines is unclear to me.

---

### Additional comment

Using the same symbol "C" for several universal constants can be confusing, as it obscures how individual lemmas relate to others in the overall argument. For instance, while the authors state that Lemma 4 can be proved by replacing part of the proof of Zhao et al. (2023) with Lemma 5, it remains unclear how the constants $c_l$ and $c_u$ influence $C$. Providing a clearer distinction between constants would improve readability and logical transparency.

---
### Minor comments

1. $M$ is undefined in Table 1: its definition first appears in Section 6.3.
2. typo in Line 855: "$\ne =$" should be corrected.
3. In definition 4, $M^{(i)}_{S_1(i)}$ appears without introduction. It seems to refer $M^{(i)}(S_1(i))$. It would be better to define the notation explicitly or ensure consistency.
4. type in Line 909: the subscript in $\theta$ is missing.

**Details Of Ethics Concerns:**

None.

---

> ### Author Response · Authors · 2025-12-04
>
> >**Weakness 1 and Question 2:** Computation cost.
>
> **Response:** We appreciate the reviewer’s concern. HOPE’s primary advantage is statistical, and in our high-dimensional setting, the sample size $N$ is significantly smaller than the ambient dimension $p$, ensuring lightweight exploitation. Following the reviewer’s suggestion, we measured runtime and found HOPE comparable to EtC baselines. We will include these results in the revision.
>
> -----
>
> >**Weaknesses 2 and Question 3:**  Comparison in Scenario 2, constants $a, b, c$
>
> **Response:**
> Thank you for highlighting this issue. We clarify the roles of the constants and improve comparison to prior work.
>
> 1. Interpretation of Examples 1(A) and 1(B).
>
> - Example 1(A): Under $\log p \le \tfrac{1}{2} T^{a} \log T$, HOPE improves upon Komiyama & Imaizumi (2024).
> - Example 1(B): For $b < 1/2$, HOPE improves upon Komiyama & Imaizumi (2024).
>
> 2. Constants $a, b, c$.
>
> - The parameters $a$ and $b$ characterize the eigenvalue decay patterns in Examples 1(A) and 1(B).
> - $c$ is not a model parameter. It is merely a symbolic placeholder denoting the coefficient in the exponent of \(p\) in the regret expression.
> We will rewrite lines 393–396 to explicitly show this and avoid confusion.
>
> -----
>
> >**Question 1:** Advantages in Scenario 4
>
> **Response:** Thanks for the thoughtful question. A simple LASSO/RDL split assumes knowledge of each arm’s structure, which is not available in Scenario 4.
> 1. Estimator selection challenge:
> Misclassifying arm structure leads to linear regret. Existing work doesn’t resolve this.
> 2. HOPE sidesteps model selection via PWE, which integrates both sparsity types and ensures sublinear regret under Assumption2.
> 3. Efficiency:
> PWE reduces estimation to scalar weighting, lowering variance and adapting to available structure.
> 5. Contribution of Scenario 4.
> This setting with mixed sparsity isn’t addressed by prior methods. HOPE is the first to offer theoretical guarantees here.
> We will revise the manuscript to emphasize these points.
>
>
> -----
> >**Question 4:** Could the authors elaborate on Lines $849-850$?
>
> **Response to Question 4:** Thanks for pointing this out. We clarify the construction of $\Gamma_t^{(i)}$ and $\bar{\zeta}$.
>
> (1) Use of the initial estimator.
> In Eq. (2), $\mathbf{Z}_t^{(i)}=\sqrt{N}\,[\,\boldsymbol{z}_t^{(i)},\,\Gamma_t^{(i)}\,]$. The definition of $\Gamma_t^{(i)}$ (Line 843) 	​ depends on the support estimator and the chosen initial estimator (e.g., LASSO or RDL).
>
> (2) Why replace one column with $\bar{\zeta}$.
> If $\theta$ were known exactly, one can show that $\beta=\Gamma(\theta)^{-1}\zeta_\theta \propto e_1$, so $\beta$ would be sparse and no additional information would be needed. When $\hat{\theta}$ is accurate, replacing a column with $\bar{\zeta}$ preserves this sparsity structure. When $\hat{\theta}$ is inaccurate (e.g., weak sparsity or $p\gg n$), the eigenvector-based columns $u_{\mathrm{eg},j}$ help compensate and provide stability. Thus the two information sources complement each other.
>
> We will revise the corresponding lines to make this intention clearer.
>
> -----
>
> >**Additional comment:** Using the same symbol "C" for several universal constants can be confusing. Providing a clearer distinction between constants would improve readability and logical transparency.
>
> **Response to Additional comment:**  We appreciate the reviewer’s remark about the use of the symbol “$C$’’ for multiple universal constants. The constant $C$ in Lemma 4 can be taken to depend only on $(c_l, c_u)$ and fixed numerical constants from Zhao et al.(2024), and is independent of $(N,p,T,K)$ and other problem parameters.
>
> To make this clearer, in the revised version we will (i) use distinct symbols such as $C_0, C_1, C_2$ for different universal constants, and (ii) explicitly state when a constant $C_j$ depends only on $(c_l, c_u)$. This will clarify how Lemma 5 feeds into Lemma 4 and improve the transparency of constant dependencies in the overall argument.
>
>
> ------
>
> >**Minor comments:** 1. $M$ is undefined in Table 1: its definition first appears in Section 6.3. 2. typo in Line 855: " $\neq=$ " should be corrected. 3. In definition $4, M_{S_1(i)}^{(i)}$ appears without introduction. It seems to refer $M^{(i)}\left(S_1(i)\right)$. It would be better to define the notation explicitly or ensure consistency. 4. type in Line 909: the subscript in $\theta$ is missing.
>
> **Response to Minor Comments:**
> We sincerely thank the reviewer for carefully reading the paper and for pointing out these helpful details. We will correct all of the issues raised.

---

### Official Review · Reviewer_d9hg · 2025-10-31

**Soundness:** 3
**Presentation:** 2
**Contribution:** 2
**Rating:** 2
**Confidence:** 4

**Summary:**

This paper combines the PWE estimator of Zhao et al. (2023) and the ETC algorithm to solve the high-dimensional linear contextual bandit problem with either a sparse parameter or approximately sparse eigenvalues of the covariance matrix of the context. Especially, the proposed method can also handle cases where both sparsities are present.

**Strengths:**

The paper proposes a novel and unified method of efficiently solving both sparse linear contextual bandits and linear contextual bandits with approximately sparse covariate matrix in eigenvalues.

**Weaknesses:**

1. The paper lacks motivation explaining why we need a better solution for both sparsities. For instance, I authors could have claimed that it is common for both sparsities to arise in practice, or while having both sparsities at once means that both LASSO and RDL can achieve sublinear regret, they are suboptimal in certain sense.
To elaborate, while Scenario 3 (both sparsities) must be the new regime of problem instances that this algorithm tackles for the first time, I think there should be more discussion about how the proposed method improves upon using one of the previous methods such as LASSO and RDL. I see that there is improvement over LASSO-ETC when $M \ll \sqrt{s_ 0}$ in Proposition 3, however it is not clear whether that is commonly the case. Similarly, Scenario 4 (mixed sparsities) can be addressed by using both LASSO and RDL depending on the arms, and the advantage of using HOPE is not explained. Without this kind of discussion, it could be slightly misleading to claim that this paper studies the both-sparsity case for the first time, because any algorithm that exploits one of the sparsities would also achieve the same guarantees when both sparsities are present.

2. I don't think comparing the regret bounds with the existing methods is fair when Assumption 1 and additional assumptions in Appendix G are required. For Assumption 1, I am not convinced that it is a standard assumption as the authors claim. Could the authors clarify which works require these assumptions and which don't, among those mainly compared in the paper? I also think assumptions in Appendix G must be included in the main paper for clarity.

3. While the abstract and the introduction give an impression that HOPE automatically adapts to two different types of sparsity, it actually undergoes a different procedure for each case. I think clarification must be made.

4. It is hard to understand what the definition of the "approximately sparse eigenvalue of the covariance matrix" condition is. Is it Definition 5 in Appendix D.6? If so, it is not clear to me what relationship this definition has with the eigenvalues of the covariance matrix.

5. It is hard to understand the procedure of computing the PWE from the main text. What is the role of the "initial estimator" in line 203? It is not used again within Section 4. Some important steps are deferred to Appendix C, and Appendix C.2 is especially hard to follow. For the second information case, I don't understand what $\Gamma_ t^{(i)}$ should be. For the both-information case, it is hard to understand why replacing one of the column vectors is necessary, and what exactly happens when the replacement occurs.

6. The presented experimental results don't seem to add much significance. The difference between the algorithms is quite small, and the nearly linear increase of the cumulative regret suggests that learning may not have happened yet, meaning that all the algorithms require more samples to properly learn the true parameter. In addition, the fact that LinUCB is competitive with other methods suggests that the experiment setting might not be sparse enough. I understand that the main contribution lies in theoretical work and I think this is a minor weakness compared to the ones listed above.

**Questions:**

1. The proof of Lemma 5 is quite confusing. I can't see where Eqs. (18)-(20) come from and how they imply the following equations in lines 1227-1238. Could the authors clarify each step? In addition, the definition of $\theta_ {Q_ x}$ is misplaced in line 1188.

2. Is Definition 5 the condition the covariance matrix $\Sigma$ must satisfy? If so, for which scenarios? As mentioned in Weaknesses, the relationship between Eq. (13) in Definition 5 and the decaying of the eigenvalues is not clear.

3. In the main text, the noise is assumed to have variance $\sigma$. Is it not necessary for the noise to be $\sigma$-subgaussian?

4. In Appendix G.3, what is $k^* $ in Line 1450?

---

> ### Author Response · Authors · 2025-12-04
>
> >**Weakness 1:** Lack of motivation for both-sparsity scenario
>
> **Response:** Thank you for emphasizing the need to motivate the both-sparsity and mixed-sparsity setting. We clarify its relevance and HOPE’s unique advantages.
>
> 1. Practical emergence of both sparsities.
> In high-dimensional settings (e.g., genetics + neuroimaging for treatment), the feature space often shows:
>   - Parameter sparsity;
>   - Eigenvalue sparsity.
> This pattern is supported by real data (e.g., Zhao et al., 2024), making this regime realistic, not artificial.
>
> 2. Why HOPE outperforms LASSO and RDL.
>
> Switching between LASSO and RDL requires knowing the sparsity type of each arm—a strong assumption. Misclassification leads to linear regret.
>
> ----
>
> >**Weakness 2:** Regret comparison under additional assumptions
>
> **Response:**  Assumption 1 is mild in high dimensions: it requires non-negligible signal and non-pathological covariance scaling—common in standard models and in Zhao et al. (2024). It’s unfamiliar mainly because PWE is novel.
>
> Importantly, Theorem 1 does not rely on Appendix G assumptions. Those are used only to verify Proposition 1 for specific estimators.
> Overall, the advantage of HOPE lies in its generality and its ability to incorporate two types of structural information, rather than requiring a separate algorithm for each regime.
>
> HOPE’s key strength is its ability to leverage both sparsity types, unlike prior work.
>
> -----
> >**Weakness 3:** Clarify whether HOPE adapts or switches
>
> **Response:** HOPE does not switch algorithms. The core estimator–bandit reduction remains fixed (Theorem 1). The different scenarios use different initial/support estimators (hyperparameters) to match structural assumptions. This parallels prior work (e.g., EtC) choosing estimators per regime.
>
> In practice, HOPE does not require prior sparsity knowledge. A brief warm-up phase or cross-validation can guide parameter choices. It also works well in a default setting, showing robustness. We will clarify this and include tuning strategies and experiments demonstrating adaptation.
>
> -----
> >**Weakness 4 and Question 2:** Clarification on Definition 5 and eigenvalue sparsity.
>
> **Response:** Approximately sparse eigenvalues” refer to decay, formalized in Definition 5 (Appx. D.6). This condition aligns with prior work (e.g., Bartlett et al., 2020; Zhao et al., 2024). Both Example 1(A) and 1(B) satisfy it.
> (1) Where it’s used: Only in Proposition 3 (Scenario 3), not in Theorem 1 or Scenarios 1/2/4.
>
> (2) Intuition: Eq. (13) bounds $\tilde{H}{\min}$, derived from tail-average eigenvalues. Fast decay (e.g., polynomial) ensures $\tilde{H}{\min}$ is small. If eigenvalues don’t decay, $\tilde{H}_{\min}$ remains large, violating the condition. Thus, Eq. (13) quantifies spectral sparsity.
>
> We will clarify this relationship, and explicitly note that Example 1(A) and 1(B) satisfy Definition 5.
>
> -----
>
> >**Weakness 5:** PWE and role of initial estimator
>
> **Response:
> (1) PWE Idea.  Eq. (1) reframes estimation as a scalar $\alpha$, with $\Gamma_t^{(i)}$ constructed to make the nuisance term $\zeta$ sparse.
>
> (2) Initial estimator’s role: Enters through $\Gamma_t^{(i)}$ (Line 843), which uses different estimators (e.g., LASSO, RDL) across scenarios. Thm. 1 applies to any that meet Assumptions 1–2.
>
> (3) Column replacement: If $\hat{\theta}$ is accurate, then $\zeta_{\hat{\theta}}$ is sparse. If not, the eigenvector columns in $\Gamma$ help recover signal via spectral structure. This design enables HOPE to combine parameter-based and covariance-based information, enhancing robustness.
>
> We will revise Appendix C to clarify these steps.
>
> ----
>
> >**Weakness 6:** Empirical results
>
> **Response:**Our goal is to validate HOPE’s consistency across regimes, not to maximize separation. We agree that higher $T$ or stronger sparsity would make gains clearer. The revised version will include such settings and tune parameters to better showcase HOPE’s performance.
>
> -----
>
> >**Question 1:** Lemma 5 derivation
>
> **Response:** Here’s the derivation:
> - Eq. (18): $\theta^\top x$ is Gaussian with variance $\theta^\top\Sigma\theta \le c_2$ ⇒ standard tail bound gives the inequality.
>
> - Eq. (19): $|x|_2^2$ concentrates under bounded eigenvalues ⇒ chi-square tail bound.
>
>  - Eq. (20): From $c_1 \le \theta^\top\Sigma\theta \le \lambda_1(\Sigma)|\theta|_2^2$, we deduce $|\theta|_2 \ge \log N / \sqrt{p}$.
>
> A union bound ensures all three hold with high probability. These bounds imply the inequalities in lines 1227–1238. We'll add these steps and fix the definition of $\theta_{Q_x}$ in the revision.
>
> -----
>
> >**Questions 3:** Subgaussian noise assumption.
>
> **Response:**
> Yes, we assume subgaussian noise with variance proxy $\sigma^2$ to enable concentration bounds.
>
> ----
>
> >**Questions 4:** Definition of $k^*$.
>
> **Response:** $k^\*$ is the coherent rank of $\Sigma^{(i)}$ with $N$ samples, defined as $k^* = k^*(\Sigma^{(i)}, N)
> := \min \{ k \in \mathbb{N}\cup\{0\} : r_k(\Sigma^{(i)}) \ge N \}$.

---

### Official Review · Reviewer_mvdW · 2025-11-01

**Soundness:** 1
**Presentation:** 1
**Contribution:** 1
**Rating:** 2
**Confidence:** 5

**Summary:**

This paper aims to unify two distinct notions of sparsity: parameter sparsity in the model itself and contextual sparsity, where the covariance matrix displays sparsity in its eigenvalues. To address this, the authors introduce a new algorithm, HOPE, which they argue outperforms existing methods when only one sparsity type is present, while also serving as a versatile approach capable of handling both sparsity regimes effectively.

**Strengths:**

The overall setup and research direction are fairly intriguing. While most studies on model parameter sparsity rely on some form of regularity assumption—such as imposing a lower bound on the smallest eigenvalue of the covariance matrix, a compatibility constant, or certain conditions on the context distribution (e.g., compatibility or margin conditions)—this paper distinguishes itself by proposing an algorithm that remains effective even when the covariance matrix exhibits eigenvalue sparsity. More importantly, it claims to function well across both sparsity regimes. Notably, Assumption 1 does not include any constraint on the minimum eigenvalue. If the theoretical results indeed hold without such assumptions, the paper’s contribution would be quite significant.

**Weaknesses:**

I have strong doubts about the soundness of this paper, especially about eigenvalues and dimensional dependencies.

It seems that the authors have likely received similar criticism before, as evidenced by the lengthy paragraph in Remark 7. However, upon close inspection, I found that the paper still hides potentially dimension-dependent quantities behind some constants.

** Regarding Assumption 2 **

The authors state that the relevant quantity is bounded by a constant $c_1$, and they refer the reader to Appendix G for verification. My reasoning is as follows:

- In Proposition 12, the error (involving $\phi$) is of order $O!\left(\sqrt{\frac{s_0 \log(p)}{\kappa \cdot N}}\right)$. (For my own reference: $d(\cdot, \cdot)$ denotes the prediction error, as defined in Definition 1, Line 883.) If this expression seems suspicious, see Section 2.4 of Bühlmann & Van de Geer (2011), Eq. (2.8).
- In Assumption 5, the constant $C$ likely hides a restricted eigenvalue. The authors may have intended to refer to Section 2.5 rather than Section 2.4 of Bühlmann & Van de Geer (2011), specifically Eq. (2.13), where the minimum signal $a_n$ appears together with the compatibility constant–based error bound. (The cited work uses the $\ell_1$-norm version, while Proposition 12 uses the $\ell_2$-norm version—corresponding to restricted eigenvalue—but in both cases, these are eigenvalue-related quantities that are being treated as constants.)
- Next, let us see how these constants affect the results. According to Proposition 6 on the PWE estimator, its error is proportional to the Lasso estimator’s error $d(\cdot)$. Thus, $\kappa$ (or the compatibility constant) enters inversely into the PWE estimator’s error bound. Tracing through the regret analysis confirms that the regret is indeed $\kappa$-dependent.
- One might argue that this dependence is not an issue if the compatibility constant or restricted eigenvalue is constant. However, that is a special case—in many realistic settings, these quantities are dimension dependent. For example, let $x^{(i)}$ be a $p$-dimensional vector drawn uniformly from the unit sphere. Its restricted eigenvalue or compatibility constant is closely related to the minimum eigenvalue, which is of order $1/p$. Hence, the authors are effectively treating a quantity that could scale with $p$ as a constant, disguising this issue under Assumption 2. For Assumption 2 to hold, one would need an unusually well-conditioned setting, such as $X \sim N(0, I)$. The authors should explicitly clarify this point.

In summary, the claim made in Remark 7—that the results are free from restricted eigenvalue or compatibility constant assumptions—is false. The authors are concealing unfavorable dependencies behind constant notation. This alone should be grounds for rejection. This issue is particularly serious because the authors explicitly state in Related Works, Line 107:

>This work, instead, focuses on the high-dimensional setting… with the feature dimension $p$ at least on the same order as the budget $T$, i.e., $p \gtrsim T$.

In such a setting, $p$-dependence is critically important.

Furthermore, I now question whether Assumption 2 itself is even correct. Shouldn’t it involve $|\hat{\theta} - \theta|_{\Sigma}$? I could not find any such term, $|\hat{\theta} \Sigma \hat{\theta} - \theta \Sigma \theta|$-ish thing in Appendix G, so clarification is needed.


Although I have not examined the sparse covariance part as carefully, even from the sparsity component alone, it is evident that the authors have made serious mathematical misstatements and overstated claims.
Therefore, I firmly believe this paper should be rejected.

**Questions:**

Please check the weaknesses above. Here are some suggestions:

1) Minimum signal condition: The minimum signal condition is far from trivial. Assumption 5 is too crucial to be buried in the appendix; its presence determines whether the convergence rate is $\sqrt{T}$ or $T^{2/3}$, as seen in Hao et al. (2020) and Jang et al. (2022). This assumption should be explicitly presented in the main text. The same goes for Assumption 4.

2) Compatibility constant assumption: Same as above, it is not trivial. Please add it in your main body.

3) In Assumption 3, what is $C_1$? While $c_1$ appeared earlier in Assumption 1, this is the first appearance of $C_1$. Is it another constant?

---

> ### Author Response · Authors · 2025-12-04
>
> >**Wekaness and Question 2:** About Assumption 2 and compatibility constant assumption
>
> **Response to Weakness and Question 2:** Thanks for bringing up these points.
> (1) Role of Assumption 2 and its relation to Appendix G.
> The main contribution of the paper is a *general* method(HOPE) for contextual bandit algorithm . Theorem 1 holds for any initial estimator satisfying Assumptions 2–3, making our method estimator-agnostic. Assumption 2 is a high-level condition on the quadratic form difference, which is central to our PWE regret analysis. Appendix G is not an additional assumption but provides examples showing how standard estimators like Lasso and RDL satisfy Assumptions 2–3 under typical high-dimensional settings.
>
> (2) On uniform-sphere example.
> This example involves extremely ill-conditioned design matrices under which many Lasso-based methods fail—including baselines we benchmark against. Our goal is not to cover such adversarial cases, but to focus on realistic regimes with exploitable sparsity or spectral decay. Theorem 1 remains valid regardless, as it does not assume any specific estimator, only that Assumption 2 is met.
>
> (3) On the form of Assumption 2.
> While Assumption 2 could be written using prediction error terms (e.g., $|\hat\theta - \theta|_\Sigma^2$), we formulate it directly in terms of the quadratic form difference since this is the natural quantity appearing in the PWE regret analysis. Appendix G shows that this bound follows from standard prediction error guarantees. We also emphasize that Appendix G derives a small prediction-error bound, whereas Assumption 2 only requires a constant-order bound—deriving such a bound from the convergence rate in Appendix G is straightforward. We will make this derivation explicit in the revised version.
>
> Lastly, regarding Remark 7, we clarify that our results do rely on compatibility/restricted eigenvalue-type assumptions when verifying Lasso’s prediction error (via Assumption 4), but these are not new or stronger assumptions—they are inherited from the estimator used.
>
> In summary, the structural conditions (including eigenvalue/compatibility–type assumptions) enter only through the verification that concrete initial estimators (such as Lasso or RDL) satisfy Assumptions 2–3. The main regret theorem for HOPE is deliberately stated in a way that is agnostic to the particular estimator and can accommodate other choices tailored to different covariance structures, including those beyond the examples discussed in the paper.
>
> -----
> >**Question 1:** About Minimum signal condition. Relationship with Hao et al. (2020) and Jang et al. (2022).
>
> **Response to Question 1:**
> (1) Hao et al. (2020) and Jang et al. (2022):
> We respectfully highlight key distinctions in both problem setting and research focus:
>
> (a) Setting difference: Hao et al. and Jang et al. primarily analyze standard linear bandits characterized by a finite number of arms and homogeneous (shared) parameters. In contrast, our work addresses contextual bandits featuring heterogeneous parameters across arms (as defined in Section 2 and further discussed in App. B). This represents a fundamentally different problem structure.
>
> (b) Focus difference: Their works establish lower bounds demonstrating the necessity of dependencies on compatibility constants ( $\kappa$ ) or eigenvalues for estimating the full parameter vector $\theta$. Our contribution centers on achieving dimension-free regret without full $\theta$ estimation, leveraging the novel mechanism of PWE.
>
> Appendix B provides a detailed comparison of these settings. To enhance reader understanding and prevent potential confusion, we will expand the paragraph titled "(Potentially) Infinite Homogeneous Arms, Fixed Heterogeneous Contexts," to explicitly contrast our methodological approach and theoretical results with those of Hao et al. and Jang et al.
>
> (2) Minimum condition
> Although Hao et al. and Jang et al. achieve $\sqrt{T}$ regret under a minimum signal condition, their setting is a strict special case of ours. For example, if our contexts were fixed and arm parameters shared, our model would reduce to theirs. Our more general setting includes time-varying contexts and heterogeneous arms. Thus, not attaining $\sqrt{T}$ regret under Assumption G.2 does not contradict the optimality of our result—it reflects a broader, more challenging setup.
>
> Therefore, while those works obtain a tighter $\sqrt{T}$ regret bound, it is achieved under much stronger and more restrictive assumptions. The fact that we do not attain $\sqrt{T}$ regret under Assumption G.2 does not contradict the optimality of our results within this broader and more challenging setting.
>
> ----
> >**Question 3:**  In Assumption 3 , what is $C_1$ ?
>
> **Response to Question 3:** $C_1$ is a separate positive constant, independent of $c_1$.

---

### Official Review · Reviewer_o1J7 · 2025-11-03

**Soundness:** 3
**Presentation:** 2
**Contribution:** 3
**Rating:** 6
**Confidence:** 1

**Summary:**

This paper focuses on high-dimensional linear contextual bandit problems. Specifically, the authors consider either sparsity in the model parameters or sparsity in the eigenvalues of the context covariance matrices. To handle both forms of sparsity, a pointwise estimator is proposed. The authors then provide regret bounds for an algorithm based on this estimator in homogeneous and heterogeneous scenarios. Finally, the superior performance of the proposed method is demonstrated through experiments.

**Strengths:**

- The paper is well-structured and easy to follow. The motivation and contributions are clearly highlighted.
- It is surprising that sublinear regret bounds can be achieved across various scenarios.
- The effectiveness of the proposed method is demonstrated not only theoretically but also empirically.

**Weaknesses:**

I am not an expert in high-dimensional contextual bandits, so I am not sure whether the provided theoretical results are stronger than those in prior work. Specifically, I wonder whether the assumptions in Propositions 1–4 are the same as or weaker than those in previous studies.

Moreover, I am curious about the relationship between parameter sparsity and eigenvalue sparsity. If the assumption on parameter sparsity holds, does eigenvalue sparsity also follow? Is one of these assumptions stronger than the other?

**Questions:**

My main concerns and questions are outlined in the Weaknesses section. Additionally, I have the following question:

- In Algorithm 1, which line uses the initial estimator?

---

> ### Author Response · Authors · 2025-12-04
>
> >**Weakness:** I am unsure whether the theoretical results in Propositions 1–4 rely on stronger assumptions than prior work. I also wonder about the relationship between parameter sparsity and eigenvalue sparsity—does one imply the other, or is one stronger?
>
> **Response to Weakness:** We thank the reviewer for raising these important questions. We clarify below the relationship between our assumptions, prior work, and the two types of sparsity considered in the paper.
>
> (1) Are our assumptions stronger than those in prior work?
> Our goal is to design a “best-of-all-worlds’’ algorithm that adapts to multiple structural regimes. Theorem 1 provides a unified regret guarantee that does not rely on the structural assumptions used in Propositions 1–4. These propositions simply instantiate Theorem 1 under different choices of initial estimators and structural assumptions so that comparisons with existing results become meaningful.
>
> - **Proposition 1 (parameter-sparse setting).** The baseline here is Li et al.(2022). Our assumptions are comparable to theirs, except for an additional mild technical condition (Assumption 5) that is used only to verify the performance of a particular support estimator. Notably, this condition does not benefit Li et al.(2022); it is introduced solely to analyze our chosen estimator within the unified framework of Theorem 1. Under these comparable assumptions, HOPE achieves regret of the same order as the existing method.
> - **Proposition 2 (eigenvalue-sparse setting).** The baseline is Komiyama and Imaizumi (2024). The assumptions required for HOPE match those used in their analysis, and HOPE improves the regret rate by leveraging PWE.
> - **Proposition 3 (both parameter and eigenvalue sparsity).** This setting is new. No prior work obtains regret guarantees here. Under the combined structure, HOPE can exploit both sources of sparsity simultaneously and achieves improved regret.
> - **Proposition 4 (mixed setting).**  In this scenario, existing single-structure algorithms incur linear regret. HOPE remains effective because it can rely on complementary sources of information, allowing it to adapt even when neither parameter sparsity nor spectral sparsity alone is sufficient.
> Thus, our assumptions are never strictly stronger than those in the corresponding prior work, and in several settings they allow us to obtain improved regret bounds.
>
> (2) Relationship between parameter sparsity and eigenvalue sparsity.
> These are two fundamentally different structural conditions:
> - **Parameter sparsity** concerns the sparsity of the regression vector $\theta$.
> - **Eigenvalue sparsity (spectral sparsity)** concerns the decay of the eigenvalues of the covariance matrix $\Sigma$.
> Neither assumption implies the other. A parameter-sparse $\theta$ does not force $\Sigma$ to have decaying eigenvalues, and a covariance matrix with fast eigenvalue decay does not imply sparsity in $\theta$. They capture orthogonal forms of structure in the model.
> This distinction explains why existing algorithms typically exploit only one type of sparsity, while HOPE can incorporate both simultaneously through the PWE construction. When either structure is present (or both), HOPE can take advantage of it; when only one holds, HOPE reduces to the best known regret for that regime.
>
> We will clarify these points in the revised version.
>
> -----
>
> >**Question:** In Algorithm 1, which line uses the initial estimator
>
> **Response to Question:
> In Algorithm 1, the initial estimator is used in Step 3 through Eq. (2), where it enters the construction of the matrix $\mathbf{Z}_t^{(i)} = \sqrt{N}\,[\,\mathbf{z}_t^{(i)},\,\Gamma_t^{(i)}\,]$. The definition of $\mathbf{Z}_t^{(i)}$ appears on Line 246, and $\Gamma_t^{(i)}$ is defined on Line 843. One of the columns of $\Gamma_t^{(i)}$ is computed using the initial estimator, which is precisely how the initial estimator is incorporated into the algorithm.
>
> For further details on the role of the initial estimator and its use in constructing the PWE, we refer the reader to Appendix C.2 of this paper, the work of Zhao et al.(2024), as well as our response to Weakness 5 raised by Reviewer d9hg.
>
> -----
> Li et al.(2022) A simple unified framework for high dimensional bandit problems. ICML.
>
> Komiyama and Imaizumi (2024) High-dimensional contextual bandit problem without sparsity. NeurIPS.
>
> Zhao et al.(2024) Estimation of linear functionals in high-dimensional linear models: From sparsity to nonsparsity. JASA.

---

### Meta-Review · Area_Chair_bTNY · 2026-01-06

**Summary:**

This paper focuses on a bandit problem with high-dimensional contexts exhibiting sparsity and introduces an algorithm that can adapt to multiple forms of sparsity. Under this framework, the authors show improved regret upper bounds in heterogeneous settings and argue that the proposed method has the ability to adapt to complex environments.

Reviewer o1J7, approaching the paper from a somewhat different area of expertise, questioned whether the assumptions introduced for the theoretical analysis are better or worse than those used in prior work, and expressed confusion about the relationship between the two types of sparsity considered in the paper. The authors responded that their assumptions are identical to those adopted in prior studies (Li et al., 2022; Komiyama and Imaizumi, 2024), and clarified that the two types of sparsity correspond to fundamentally different assumptions. From an expected reviewer reaction, these responses are clear and precise and would likely resolve Reviewer o1J7’s concerns.

Reviewer mvdW argued that Assumption 2, used in the regret analysis, involves a constant that depends on the eigenvalues of the context distribution, which can become very large in certain scenarios. On this basis, the reviewer strongly claimed that several statements in the paper are overstated. The reviewer also emphasized the importance of the minimum signal condition. The authors replied that Assumption 2 is required to apply an existing estimator (the Lasso) used within their method and does not directly affect the core novelty of the paper. Regarding the minimum signal condition, they highlighted the differences from prior work and argued that their formulation captures a more general setting. While this reviewer appears to maintain a rather strong opposition, the argument that Assumption 2 is inherited from existing methodology and is not intrinsic to the novel component of the proposed approach is reasonably persuasive.

Reviewer d9hg questioned both the importance of considering two types of sparsity and the naturalness of simultaneously addressing them. The reviewer also pointed out the strength of some assumptions, shortcomings in presentation, and the relatively small empirical improvements observed in the experiments. The authors provided brief responses to these points. However, from an external perspective, the responses appear to largely restate what is already written in the paper rather than directly engaging with the reviewer’s underlying concerns. Under these circumstances, it would be difficult for the reviewer to conclude that their concerns had been resolved.

Reviewer 6few raised concerns about potentially large computational costs associated with the algorithm’s implementation, as well as unclear presentation of certain constants and aspects of the algorithmic design. The authors argued that the computational cost is not excessive in practice and clarified several constants and design choices, stating that they would improve the presentation. From an anticipated reviewer reaction, while experimental results suggest comparable runtimes in practice, a fundamental limitation, namely, the need to recompute estimators separately for each dimension, does not appear to have been directly addressed. Consequently, it is not obvious that the computational cost issue is fully resolved in general, and more comprehensive experiments or more convincing theoretical arguments would be needed. With respect to presentation, while explanations were provided on a case-by-case basis, some of the reviewer’s questions concern the paper as a whole, and it is unclear whether the responses fully address those broader concerns.

Taking all of the above into account, it is reasonable to expect that at least two reviewers would still feel that their concerns have not been resolved. Some criticisms. such as the strength of certain assumptions, originate from limitations inherent in existing methods rather than from this paper itself. However, the more fundamental issue of simultaneously handling two types of sparsity does not appear to have been explained in a way that convincingly addresses the reviewers’ doubts.

**Reviewer Concerns:**

See above.

**Reviewer Scores:**

See above.

---

### Decision · Program_Chairs · 2026-01-26

Reject